# NOS inhibition sensitizes metaplastic breast cancer to PI3K inhibition and taxane therapy via c-JUN repression

Tejaswini Reddy[1,2,3,11], Akshjot Puri[2,3,11], Liliana Guzman-Rojas[2], Christoforos Thomas [2,3], Wei Qian[2], Jianying Zhou[2], Hong Zhao [2,3], Bijan Mahboubi[4], Adrian Oo[5], Young-Jae Cho[5], Baek Kim [5], Jose Thaiparambil[2], Roberto Rosato [2], Karina Ortega Martinez[2], Maria Florencia Chervo[2], Camila Ayerbe[6], Noah Giese [2,3], David Wink [7], Stephen Lockett[8], Stephen Wong [2,3], Jeffrey Chang [6,9], Savitri Krishnamurthy[9], Clinton Yam [9], Stacy Moulder[10], Helen Piwnica-Worms [9], Funda Meric-Bernstam [9] & Jenny Chang[2,3] ✉

Metaplastic breast cancer (MpBC) is a highly chemoresistant subtype of breast cancer with no standardized therapy options. A clinical study in anthracycline-refractory MpBC patients suggested that nitric oxide synthase (NOS) inhibitor NG-monomethyl-l-arginine (L-NMMA) may augment anti-tumor efficacy of taxane. We report that NOS blockade potentiated response of human MpBC cell lines and tumors to phosphoinositide 3-kinase (PI3K) inhibitor alpelisib and taxane. Mechanistically, NOS blockade leads to a decrease in the S-nitrosylation of c-Jun $NH_2$-terminal kinase (JNK)/c-Jun complex to repress its transcriptional output, leading to enhanced tumor differentiation and associated chemosensitivity. As a result, combined NOS and PI3K inhibition with taxane targets MpBC stem cells and improves survival in patient-derived xenograft models relative to single-/dual-agent therapy. Similarly, biopsies from MpBC tumors that responded to L-NMMA+taxane therapy showed a post-treatment reversal of epithelial-to-mesenchymal transition and decreased stemness. Our findings suggest that combined inhibition of iNOS and PI3K is a unique strategy to decrease chemoresistance and improve clinical outcomes in MpBC.

Metaplastic breast cancer (MpBC) is a rare and highly aggressive malignancy that exhibits the most dismal prognosis of all breast cancers, with a survival rate of 8 months or less in patients with metastatic disease[1]. MpBCs are typically triple-negative, lacking the expression of estrogen receptor (ER), progesterone receptor (PR), and human epidermal growth factor receptor 2 (HER2), and histologically characterized by the presence of both epithelial and mesenchymal components[2,3]. Patients with MpBC receive similar therapeutic

[1]Department of Internal Medicine, Baylor College of Medicine, Houston, TX, USA. [2]Houston Methodist Research Institute, Houston, TX, USA. [3]Houston Methodist Neal Cancer Center, Houston, TX, USA. [4]Adams School of Dentistry, University of North Carolina, Chapel Hill, USA. [5]Department of Pediatrics, School of Medicine, Emory University, Atlanta, GA, USA. [6]McGovern Medical School, The University of Texas Health Science Center, Houston, TX, USA. [7]Cancer Innovation Laboratory, Center for Cancer Research, National Cancer Institute, National Institute of Health, Frederick, MD, USA. [8]Optical Microscopy and Analysis Laboratory, Cancer Research Technology Program, Frederick National Laboratory for Cancer Research, National Cancer Institute, National Institutes of Health, Frederick, MD, USA. [9]The University of Texas MD Anderson Cancer Center, Houston, TX, USA. [10]Eli Lilly and Company, Indianapolis, IN, USA. [11]These authors contributed equally: Tejaswini Reddy, Akshjot Puri. ✉e-mail: jcchang@houstonmethodist.org

interventions to patients with non-metaplastic triple-negative breast cancers (non-MpTNBC); however, MpBC tumors are more chemoresistant and have a worse overall survival (OS) than non-MpTNBC[4,5].

Treating MpBCs remains a therapeutic dilemma due to its rarity (<5% of all breast cancers) and our limited understanding of its pathogenesis[5]. However, recent multi-omic analyses of MpBC tumors have revealed molecular aberrations/characterizations that may be therapeutically targeted[5]. For example, MpBC tumors are enriched in epithelial-to-mesenchymal transition (EMT)/cancer stem cell (CSC) features, and predominately have aberrations in both the PI3K and iNOS pathways[6–9]. We discovered a new cancer gene, ribosomal protein L39 (RPL39), that is associated with therapy resistance, CSC self-renewal, and lung metastases in TNBC and MpBCs[8,10]. Mechanistically, RPL39 is reported to promote the production of nitric oxide (NO) by regulating the function of iNOS[8]. Elevated expression of RPL39 and iNOS in MpBC tumors are poor prognostic indicators and are associated with worse OS[8].

We previously demonstrated that NOS inhibition with the pan-NOS inhibitor NG-monomethyl-L-arginine (L-NMMA) decreased tumor cell proliferation, mammosphere formation, and migration in vitro and reduced tumor development and growth as well as lung metastasis in TNBC patient-derived xenograft (PDX) models[11,12]. These preclinical studies provided the rationale for conducting a phase I/II clinical trial of L-NMMA plus taxane for treating patients with chemorefractory, locally advanced breast cancer (LABC) or metastatic TNBC (NCT02834403). In that study, the overall response rate (ORR) of patients was 45.8% (81.8% for LABC, 9/11), with no grade ≥3 toxicities attributed to L-NMMA[13]. These findings suggested that targeting NOS may be a safe and effective therapeutic target for chemorefractory breast cancers.

Furthermore, NO can uniquely activate multiple oncogenic signaling pathways, including those mediated by PI3K, extracellular signal-regulated kinase, β-catenin, transforming growth factor beta (TGFβ), and hypoxia-inducible factor (HIF)[14,15]. Considering that MpBC produces high levels of NO that can activate PI3K signaling and harbors aberrations in the PI3K pathway, we assessed a combinatorial approach of co-targeting NOS and PI3K/Akt, using the FDA approved α-specific PI3K inhibitor alpelisib (Piqray®, Novartis) and pan-NOS inhibitor L-NMMA, as an effective therapeutic strategy.

In this work, we use patient-derived MpBC cell lines and PDX models to demonstrate that iNOS inhibition augments the efficacy of PI3K inhibitor therapy against MpBC. This combinatorial effect is mediated in part through NOS inhibition inducing a reversal of EMT programming in MpBC cells, leading to enhanced chemosensitivity. Our findings suggest that combining NOS and PI3K inhibition is an effective therapeutic strategy to treat MpBC by reversing EMT and decreasing CSCs, rendering MpBC tumors more chemosensitive.

## Results
### Evidence of clinical activity of NOS inhibition with L-NMMA combined with taxane in MpBC patients
We first discovered the clinical relevance of targeting the NOS signaling pathway in MpBC when retrospectively evaluating the results of a phase I/II clinical trial of L-NMMA plus taxane for treating patients with chemorefractory, LABC, or metastatic TNBC[13]. In this trial, 35 TNBC patients were recruited, and 15 patients had MpBC (Phase 1B, n = 4; Phase 2, n = 11); 86.6% (13/15) patients had metastatic breast cancer (MBC), with a median of 2 prior lines of therapy (range 0–5) and 13.3% (2/15) had anthracycline-refractory LABC (Fig. 1A, Supp. Fig. 1A). The median age of the cohort was 62 years (35–75 years). Two patients were excluded from analysis due to adverse events unrelated to treatment. The clinical benefit rate (CBR) was 46% (6/13); the ORR was 23% (3/13) with one partial response (PR) in metastatic TNBC, one pathological complete response (pCR) and one PR in LABC (Fig. 1A, Supp. Fig. 1A). Grade 3 or more toxicity was seen in 15% (2/13) patients;

however, none were attributed to L-NMMA and were likely related to the standard of care docetaxel. The median progression-free survival and median OS for MBC patients were 4.5 months (range 3–7 months) and 12.8 months, respectively. The clinical characteristics of all patients are listed in Supplementary Tables 1 and 2.

The patient with pCR at the time of trial enrollment had a bulky primary mass with extensive skin ulceration which was refractory to anthracycline and achieved a complete response after L-NMMA therapy (Fig. 1B–D) and showed the highest intensity of iNOS expression at baseline (BL) (Fig. 1A, D, E). For all responders, iNOS expression had a significant reduction at end-of-treatment (EOT), underscoring the importance of NOS inhibition in combined therapy for improving clinical response. In three out of six non-responders, iNOS expression increased from BL to EOT with statistical significance in two non-responders (p = 0.002, p < 0.001) (Fig. 1E). Since MpBC tumors typically have genomic alterations in PI3K signaling along with increased iNOS activation, we also evaluated available next-generation sequencing data of TP53, PIK3CA, PIK3R1, and AKT1 mutations for 7/13 patients (Supp. Fig. 1B). 4/7 patients had TP53 mutations, one with progressive disease (PD), two with stable disease (SD), and one with a partial response. 2/7 patients had PIK3CA mutations and were non-responders with disease progression. Two patients with SD had AKT1 mutations, and one with a concomitant PIK3R1 mutation. Next generation sequencing for 6/13 patients was not available because either the analysis was not conducted or because the data was not available to us for further analysis.

These clinical findings suggested that targeting iNOS may be a potential therapeutic strategy to augment the efficacy of taxane and potentially targeted therapies in chemoresistant MpBC. To fully define the relevance of targeting iNOS in MpBC, we analyzed available transcriptomic data from The Cancer Genome Atlas (TCGA) cBioPortal Combined Dataset (n = 179,290) (cbioportal.org), which showed that 'MpBC' was the top breast cancer subtype and 4th highest cancer to harbor NOS2 genomic aberrations (Fig. 1F). TCGA is limited in the number of MpBC cases and associated clinicopathological characteristics, thereby making it difficult to evaluate any potential clinical correlations with NOS expression. To further understand the clinical importance of NOS in MpBC, we next analyzed datasets from the ARTEMIS clinical trial (NCT02276443) that recruited patients with newly diagnosed TNBC and MpBC, and provided us an avenue to investigate the genomic landscape of patients with these cancers[16]. Transcriptomic RNA-seq analysis of treatment-naïve TNBC tumors enriched with MpBC histology and metaplastic-like characteristics from ARTEMIS found that MpBC+metaplastic-like tumors had higher tumor NOS2 expression than non-MpTNBC (Fig. 1G), further supporting our findings from the TCGA cBioPortal Combined Dataset (Fig. 1F). More significantly, high tumor NOS2 mRNA expression was correlated with significantly worse metastasis-free survival (MFS) [p = 0.042], indicating the importance of inhibiting iNOS to improve clinical outcome in patients with MpBC (Fig. 1H).

We next compared the transcriptomes of MpBC to invasive ductal carcinoma (IDC) tumors from TCGA (Fig. 1I, Supp. Fig. 1C). Relative to IDC, the top enriched pathways in MpBC tumors include EMT, hypoxia, PI3K-Akt signaling pathway, and extracellular matrix regulation (Fig. 1I). These findings further support increased signaling of iNOS in MpBC considering the established link between HIF-1α-induced hypoxia and enhanced iNOS function, the proposed alteration of EMT programming by iNOS activation, and the reported stimulation of PI3K pathway by iNOS-derived NO regardless of PIK3CA mutation status[10,11,17].

### Enhanced co-activation of iNOS and PI3K signaling in MpBC tumors
To understand the biological relationship between iNOS and PI3K signaling pathways, we first queried the STRING database (http://

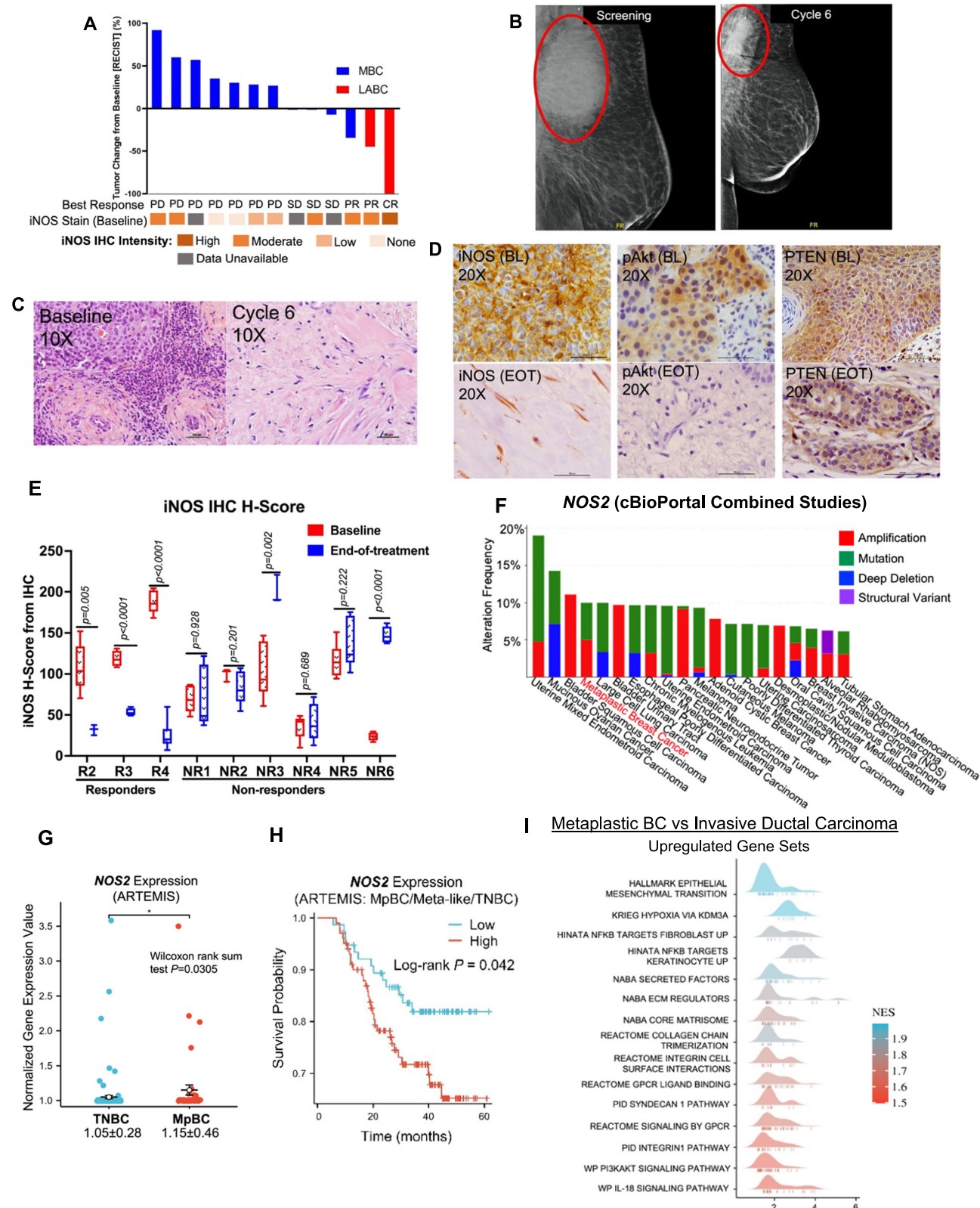

 protein-protein interactions and found that *NOS2* functionally interacts with *AKT1* with a high enrichment *p* value (0.000538) (Fig. 2A). We also identified biological interactions of NOS with EMT, PI3K, and hypoxia-related genes (*p* = 1.21e-08) (Supp. Fig. 2A) that strongly support the clinical relevance of NOS and PI3K pathways and their potential interaction in MpBC. Mutual exclusivity analysis in data from

cBioPortal Combined Studies (*n* = 179,290) and PanCancer Atlas Studies (*n* = 76,639) of TCGA[19,20] showed that genomic alterations in *NOS2* significantly co-occur with those of *PIK3CA, PTEN*, and *AKT1* genes (all *p* < 0.001) in multiple cancer types (Fig. 2B, Supp. Fig. 2B). Similar significant co-occurrence of *NOS2 and AKT1* gene alterations were found in the Breast Cancer cohort from TCGA (*n* = 8644) (*p* < 0.001), suggesting that iNOS and PI3K signaling are complementary and have

**Fig. 1 | Clinical activity of L-NMMA combined with docetaxel in chemore-fractory MpBC patients. A** Waterfall Plot showing change in target lesion/tumor volume, best response (RECIST), and baseline iNOS H-score expression status in the 13 patients with available data from post-baseline assessments. PD progressive disease, SD stable disease, PR partial response, and CR complete response. iNOS H-Score Metric (0–50: None, 51–100: low, 101–150: Moderate, >150: High) **B** Representative mammogram images at baseline (BL) and end of cycle 6 for Patient 100-048 (CR), showing shrinkage in target lesion size. **C** Hematoxylin and Eosin staining of BL tumor from 100-048 showing metaplastic squamous differentiation and end-of-treatment (EOT) tissue showing residual keratinized and fibrotic tissue. Scale bars represent 100 μM. Representative images of $n = 3$ tissue samples. **D** BL IHC staining of iNOS, PTEN, and phospho-Akt of tumor tissue from 100-048 and associated EOT iNOS, pAkt, and PTEN staining in residual tissue. Scale bars represent 100 μM. **E** iNOS H-score for BL and EOT tumors in indicated responders and non-responders. BL and EOT H-scores are indicated by red and blue box and whisker plots, respectively. R2-R4 indicates responder patients, and NR1-

NR6 indicate non-responder patients. H-score analysis of 4–6 images per slide captured from tissues to cover entire tissue bed. The center bar indicates median, bounds of box represent lower and upper quartiles, and the whiskers indicate minimum and maximum of the dataset for each group. Statistical analysis by two-sided Student's $t$ test. **F** *NOS2* alteration frequencies in various cancers from cBio-Portal Combined Study Dataset ($n = 179290$). **G** *NOS2* normalized gene expression in MpBC/metaplastic-like TNBC ($n = 41$, red dots) and non-MpTNBC ($n = 137$, blue dots) from the ARTEMIS dataset. The dot and error bars represent mean ± SD. Statistical analysis by two-sided Student's $t$ test. **H** Kaplan-Meier metastasis-free survival analysis of MpBC and non-MpTNBC tumors based on expression status [high (red line)/low (blue line)] of *NOS2* from ARTEMIS dataset. **I** Gene set enrichment analysis of the top represented upregulated hallmark gene sets based on normalized enrichment score (NES) from RNA-sequencing data from TCGA in human MpBC tumors ($n = 14$) compared to invasive ductal carcinoma tumors ($n = 814$).

collaborative oncological function (Supp. Fig. 2C). The significance of these *NOS2* genetic alterations in relation to disease pathogenesis is currently unknown; however, the significant co-occurrence between *NOS2* and PI3K signaling genes implies that these pathways may be influencing tumorigenesis by utilizing similar mechanisms in multiple cancer types, including MpBC.

Next, we assessed the distribution of iNOS and active Akt co-expression among breast cancer subtypes: ER+, HER2+, TNBC, and MpBC. We compared the expression of iNOS and phosphorylated forms of Akt (Ser473 and Thr308) in 15 breast cancer cell lines and human mammary epithelial cell line MCF-10A. Relative to non-metaplastic TNBC cell lines, MpBC cell lines had exceptionally elevated protein levels of iNOS ($p = 0.0426$) and phospho-Akt (Thr308) ($p = 0.0223$) Fig. 2C, D. Using immunohistochemistry (IHC), we evaluated the protein expression of iNOS and phospho-Akt in tissues from breast cancer PDX models and found that MpBC PDXs predominantly had more co-expression of high/moderate staining of iNOS and phospho-Akt relative to other breast cancer PDX subtypes (Fig. 2E).

Using droplet digital polymerase chain reaction (ddPCR), we also evaluated whether these corresponding breast cancer PDXs harbored the *RPL39 A14V* mutation, associated with iNOS activation[8], and *PIK3CA* hotspot mutations (*E542K, E545K, H1047R, H1047L*). The *RPL39 A14V* mutation was detected in four out of six (66%) MpBC PDX models versus one out of 21 (4.7%) TNBC PDX models ($p = 0.004$) and no mutations were found in ER+ or HER2+ models. *PIK3CA* hotspot mutations were found in 3/6 (50%) MpBC PDX models versus 4/21 (19%) TNBC, 2/5 ER+, and 2/3 HER2+PDX models (Fig. 2E, F). Specifically, the MpBC PDX models PIM-010 and PIM-084 were *RPL39/PIK3CA*-mutated and had a corresponding expression of iNOS (high) and phospho-Akt (Ser473). Consistent with the protein levels, we observed substantially higher *NOS2* mRNA expression in MpBC tumors compared to other breast cancer subtypes by performing quantitative PCR (qPCR) analysis in the corresponding 35 PDX models (Fig. 2G). Overall, these findings underscore the significance of iNOS as a molecular target and propose its inhibition alone or together with the co-activated PI3K as a potential therapeutic strategy for MpBC.

### NOS inhibition acts synergistically with PI3K inhibitor in MpBC cell lines

To explore whether a combined inhibition of iNOS by L-NMMA and PI3K signaling by alpelisib is synergistic in MpBC models, we performed a small cell line screen with MpBC cell lines BT549, Hs578T, SUM159, and HCC1806. These cell lines were derived from primary carcinosarcomas and/or exhibited metaplastic features by pathology[8,21,22] (Supp. Fig. 3A).

The IC$_{50}$ values of alpelisib in each MpBC cell line are summarized (Supp. Fig. 3B). Notably, three MpBC cell lines (SUM159, Hs578T, and BT549) were sensitive to alpelisib and achieved IC$_{50}$ concentrations

under the threshold for a physiological achievable dose (5.6 μM), based on a previous phase 1b trial[23]. Cell viability assays revealed that combining L-NMMA and alpelisib was synergistic in SUM159 and Hs578T cell lines, which harbor *PIK3CA* and *PIK3R1* mutations, respectively (combination index [CI] <1). L-NMMA and alpelisib combination therapy was antagonistic in BT549 (*PTEN* mutated) and HCC1806 (*PIK3CA/PTEN* wild-type) cell lines (CI > 1) (Fig. 3A). The synergistic effect seen in SUM159 and Hs578T cell lines was also confirmed with anchorage-dependent colony formation (Supp Fig. 3C) and caspase 3/7 activation assays (Supp. Fig. 3D). To confirm this observation, we generated clones of SUM159 cells with CRISPR/CAS9-mediated knockout of *NOS2* (NOS2KO), confirmed by the absence of iNOS protein in immunoblots and reduced production of nitrite/nitrate in NOS2KO clones relative to parental SUM159 cells (Fig. 3B, Supp. Fig. 3E). The absence of iNOS in the NOS2KO clones was associated with substantially reduced expression of phospho-Akt (Ser473/Thr308), suggesting that NOS downregulation could impair Akt activation. Further, consistent with the pharmacological inhibition, ablation of NOS in SUM159 NOS2KO clones led to enhanced sensitivity to alpelisib (Fig. 3A-C). Finally, pharmacological inhibition of NOS resulted in decreased phosphorylation of Akt and S6 in MpBC cell lines SUM159 and Hs578T, further reduced by the addition of PI3K inhibition, resembling the synergistic effect of two inhibitors on the survival of the same cell lines (Fig. 3A, D, and Supp. Fig. 3F). The combination therapy did not significantly reduce phosphorylation of Akt and S6 in BT549 cell line (Fig. 3D, Supp Fig. 3F). We also evaluated whether the differential response to NOS inhibition was associated with variability in S-nitrosoglutathione reductase (GSNOR) expression. We found that cell lines responsive to combination therapy (SUM159 and Hs578T) had a relatively reduced expression of GSNOR, compared to cell lines that were not responsive to combination therapy (HCC1806 and BT549) (Fig. 3E).

To better understand the underlying mechanism of the synergistic cytotoxic effect of NOS and PI3K inactivation on MpBC cells, we studied cellular events affected by single-agent treatment. PI3K inhibitors, such as alpelisib, induce DNA damage via nucleotide depletion and are synergistic with PARP inhibitors in *BRCA*-mutant breast cancers[24,25]. Our findings show that NOS inhibition alone induced DNA damage and the combination of L-NMMA with alpelisib augmented this effect in *PI3KCA/PIK3R1* mutated cells (SUM159/Hs578T), as shown by increased comet tail moment (Fig. 3F), enhanced γH2AX signal with decreased Rad51 foci formation in immunofluorescence analysis, and time-dependent increase in γH2AX protein levels with concomitant decrease in Rad51 expression (Supp. Fig. 3G, H). In contrast, in *PTEN*-deleted BT549 cells, NOS inhibition was unable to enhance the DNA-damaging effects of alpelisib (Fig. 3F, Supp Fig. 3I). The involvement of DNA damage in the synergistic cytotoxic effect in the presence of *PIK3CA* mutations was supported by enhanced nucleotide depletion observed in the presence of the combination therapy (L-NMMA

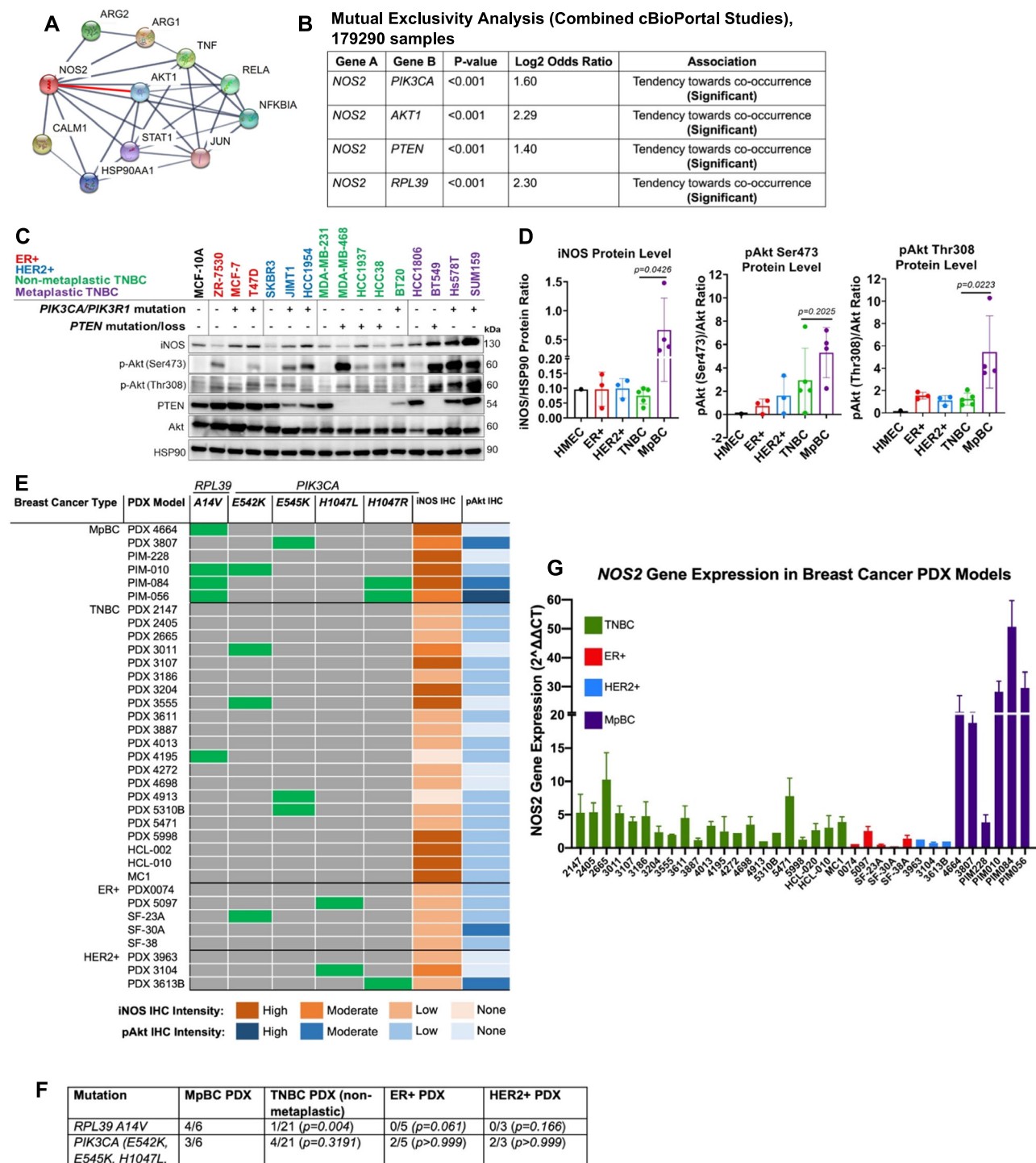

+alpelisib) in SUM159 cells but not in BT549 cells using an HIV reverse transcriptase (RT)-based dNTP assay (Supp. Fig. 3J, K). These results suggest that L-NMMA augments the effect of PI3K inhibitor in inducing DNA damage in MpBC cells with *PIK3CA* or *PIK3R1* mutations by enhanced nucleotide depletion, leading to increased apoptosis and enhanced sensitivity of MpBC cells to dual NOS and PI3K inhibition.

### NOS inhibition sensitizes MpBC tumors to PI3K inhibitor and taxane chemotherapy in vivo

To support our in vitro studies, we evaluated the effect of combined NOS and PI3K inhibitor treatment in MpBC PDX tumors. In orthotopic MpBC PDX models, mice were randomly assigned to one of four groups to receive vehicle, L-NMMA, alpelisib, or combination of L-NMMA and alpelisib (Fig. 4A). The combination of L-NMMA+alpelisib was well tolerated as indicated by mouse body weight over time (Supp. Fig. 4A, B). L-NMMA+alpelisib therapy resulted in a significant inhibition of tumor growth relative to single-agent alpelisib therapy in *PIK3CA*-mutant MpBC PDX models (BCM-3807 [$p = 0.0456$], PIM-010 [$p = 0.0079$], and PIM-084 [$p = 0.016$]) (Fig. 4B–D). In BCM-4664 (*PIK3CA*-wild-type) PDX model, the addition of L-NMMA to alpelisib did not significantly reduce tumor volume relative to single-agent alpelisib therapy ($p = 0.1275$) (Fig. 4E). However, we saw a trend toward enhanced tumor response to combination therapy in this PDX, suggesting that NOS inhibition may augment the efficacy of PI3K

**Fig. 2 | Co-expression of iNOS and phospho-Akt is predominant in MpBC.**
**A** Network of protein interactions with iNOS generated by STRING analysis. Each network node represents one gene. Red line highlight interaction between *NOS2* and *AKT1*. **B** Mutual exclusivity analysis of cBioPortal combined dataset of all cancers (*n* = 179,290) demonstrated a significant tendency toward co-occurrence for *NOS2* with *PIK3CA, AKT1, PTEN*, and *RPL39* genomic alterations. mRNA expression and protein/phosphoprotein data were selected in analysis. Statistical analysis using one-sided Fisher's Exact Test. **C** Immunoblotting analysis of iNOS, PTEN, phospho-Akt (Thr308 and Ser473) in a panel of breast cancer cell lines with known *PIK3CA/PIK3R1* and *PTEN* mutation status. HSP90 was used as a loading control. Human mammary epithelial cell line (MCF-10A), ER+ breast cancer, HER2+ breast cancer, non-metaplastic TNBC, and metaplastic TNBC cell lines are indicated in black, red, blue, green, and purple, respectively. Blots shown are representative images of *n* = 3 biological replicates. **D** Comparison of normalized iNOS and phospho-Akt (Thr308 and Ser473) protein levels among MpBC, TNBC, ER+, and

HER2+ breast cancer cell lines. Each dot represents a cell line showing a representative experiment, statistical analysis by two-sided Student's *t* test comparing differences in protein expression ratios of MpBC to TNBC cell lines. Bars and error bars represent Mean ± SD. **E** Droplet digital PCR analysis of *RPL39 A14V* and *PIK3CA* hotspot mutations (*E542, E545K, H1047L, H1047R*) and iNOS/phospho-Akt (Ser473) immunohistochemical expression status in PDXs of TNBC (*n* = 12), ER+ (*n* = 5), HER2+ (*n* = 3), and MpBC (*n* = 6). Green bars represent PDX models that express the specifically indicated mutation in the column. Blue bars (depending on gradient) represent the expression of phospho-Akt from low to high expression, a marker of PI3K/Akt activation. Orange bars (depending on gradient) represent low to high iNOS expression. **F** Results of two-sided Fisher's exact test comparing MpBC mutation status of *RPL39 A14V* and *PIK3CA* hotspot mutations in other breast cancer subtypes. **G** mRNA expression of *NOS2* in all breast cancer PDX models. Values were compared to ΔCT value from PDX 4913 (TNBC) as a control that was set to 1 and represent the mean ± SD of three biological replicates.

inhibition regardless of *PIK3CA* mutation status. Combination therapy also resulted in a significant decrease in protein levels of iNOS and phospho-Akt, as shown by quantification of IHC H-score analysis (Supp. Fig. 4C–E) and the enhanced expression of cleaved caspase 3 in BCM-3807 tissues (Supp. Fig. 4F).

Driven by the benefit of the combination therapy (L-NMMA plus taxane) in the clinical trial with the chemorefractory MpBC patients (Fig. 1) and considering that MpBCs have a worse pCR rate than non-MpTNBC to standard-of-care chemotherapeutic regimens[26], we next evaluated whether L-NMMA alone and its combination with alpelisib was effective at augmenting the efficacy of chemotherapy in chemoresistant MpBC tumors (Supp. Fig. 4G). We treated PDX models BCM-4664 (*PIK3CA* wild-type) and BCM-3807 (*PIK3CA* E545K) with three cycles of taxane +/− targeted therapies. The triple combination therapy (L-NMMA+alpelisib+docetaxel) reduced the volume of BCM-4664 tumors and significantly improved OS of recipient mice in comparison to docetaxel+alpelisib therapy (*p* = 0.0432) and docetaxel +L-NMMA therapy (*p* = 0.0007) (Fig. 4F, H). In BCM-3807 model, the combination therapy eradicated the tumors in 5 out of 7 animals with no detectable tumor re-growth after treatment discontinuation (Day 43) (Fig. 4G) and extended OS in comparison to docetaxel-only treatment arm (*p* = 0.0192) (Fig. 4I). Cell viability assays also confirm our findings showing that NOS2KO SUM159 clones were more sensitive to docetaxel treatment than control SUM159 cells. (Supp. Fig. 4H).

**NOS inhibition reverses EMT transition in MpBC cell lines and PDX models**

To define the molecular mechanism of the synergistic effect of NOS and PI3K inhibition in MpBC, we performed RNA-Seq/GSEA in MpBC PDX tumors following treatment with dual combination therapy. GSEA indicated that L-NMMA+alpelisib treated tumors were enriched with pathways associated with cellular epithelization and differentiation, including the formation of cornified envelopment, keratinization, and negatively enriched for hypoxia and EMT pathways (Fig. 5A, Supp Fig. 5A). This analysis suggested that combined NOS and PI3K inhibition may have enabled MpBC tumor cells to undergo reversal of EMT and enhanced cellular epithelization. We then conducted immunofluorescence analysis of BCM-3807 and PIM-010 treated tumors to determine expression of the epithelial marker E-cadherin and ZEB1[27]. L-NMMA single-agent and L-NMMA+alpelisib treated tumors had increased expression of E-cadherin with an associated decrease in ZEB1 (Fig. 5B, C, Supp. Fig. 5B, C), suggesting that NOS inhibition may have induced a reversal of EMT, as suggested in our previous study using TNBC models[11]. We further investigated this mechanistic relationship in inducing EMT reversal by evaluating MpBC cell lines that were specifically responsive to combined PI3K and NOS inhibition, SUM159 and Hs578T, based on findings from Fig. 3. In MpBC cell lines (Hs578T and SUM159), inhibition of NOS alone by L-NMMA and combined with the PI3K inhibitor alpelisib led to reduced protein levels of ZEB1 and

downstream CHK1, with an increased expression of E-cadherin and zonula occludens-1 (ZO-1) (Fig. 5D, Supp. Fig. 5D). Consistent with the repressive effect of NOS inhibitor on EMT, SUM159 cells with *NOS2* knockout were more compact with a cobblestone-like morphology compared with parental cells that had decreased adhesions, spindle-like structures, and enlarged protrusions (Fig. 5E). This result was further supported by decreased protein levels of iNOS, phospho-Akt, HIF-1α, Zeb1, vimentin, and latent TGFβ, increased expression of E-cadherin and ZO-1 and decreased migratory capacity of NOS2KO clones relative to parental SUM159 cells (Fig. 5F, Supp. Fig. 5E, F).

To evaluate how NOS inhibition caused reversal of EMT in MpBC cells, we performed RNA-sequencing in SUM159 NOS2KO cells. Gene ontology analysis of the transcriptomic data revealed a list of significantly downregulated genes in NOS2KO cells associated with EMT induction and CSC maintenance including *LCN2, TGFB1, GLI1, NOTCH1, COL17A1* (Fig. 5G). In addition, the NOS2KO clones had corresponding enrichment of pathways such as homophilic cell adhesion, epithelial differentiation, and extracellular matrix organization (Fig. 5H). We then assessed the expression of two of the downregulated genes and potent drivers of EMT by qPCR and immunoblotting, and found substantial decrease in the mRNA and protein levels of *TGFB1* and *LCN2* in SUM159 NOS2KO clones, indicating the involvement of iNOS in the transcriptional regulation of these genes[28,29] (Fig. 5I–K). We confirmed the central role of TGFβ in mediating the effect of NOS on EMT in MpBC by treating SUM159 NOS2KO cells with TGFβ. As expected, activation of TGFβ completely reversed the epithelial transformation induced by the downregulation of NOS in SUM159 cells, as indicated by the prevalence of mesenchymal morphology and strong upregulation of the mesenchymal markers Vimentin and ZEB1 in TGFβ-treated NOS2KO cells (Supp. Fig. 5G, H). These findings enabled us to focus on the transcriptional machinery regulating the expression of *TGFB1* and *LCN2* to identify direct targets of NOS in MpBC cells, including factors that act on AP-1 response elements like c-Jun and Fos[30].

According to our STRING database query, *NOS2* had a significant biological interaction with *JUN* (Fig. 2A), therefore we hypothesized that NO may trigger the activation of AP-1 transcription factors, leading to altered expression of EMT effectors and overall altered EMT programming. To test our hypothesis, we examined the factors necessary for the transcription of *LCN2/TGFB1* in MpBC cells by treating SUM159 cells with siRNAs specific to *CREB3, XBP1, FOS*, and *JUN* (Supp. Fig. 5I).

We observed that *JUN* knockdown resulted in the strongest associated decrease in the mRNA levels of *TGFB1* and *LCN2*, suggesting that c-Jun may be the direct mediator of the transcriptional effects of iNOS on EMT by regulating *TGFB1* and *LCN2* (Fig. 5L, M, Supp. Fig. 5I). The direct effect of iNOS on c-Jun transcriptional complex in MpBC is further supported by previous studies showing that the regulator of c-Jun, c-Jun N-terminal kinase (JNK) is subject to S-nitrosylation-mediated activation in cardiac dysfunction and fibrosis[31]. Based on our findings and these previous studies, we deduced that in MpBC

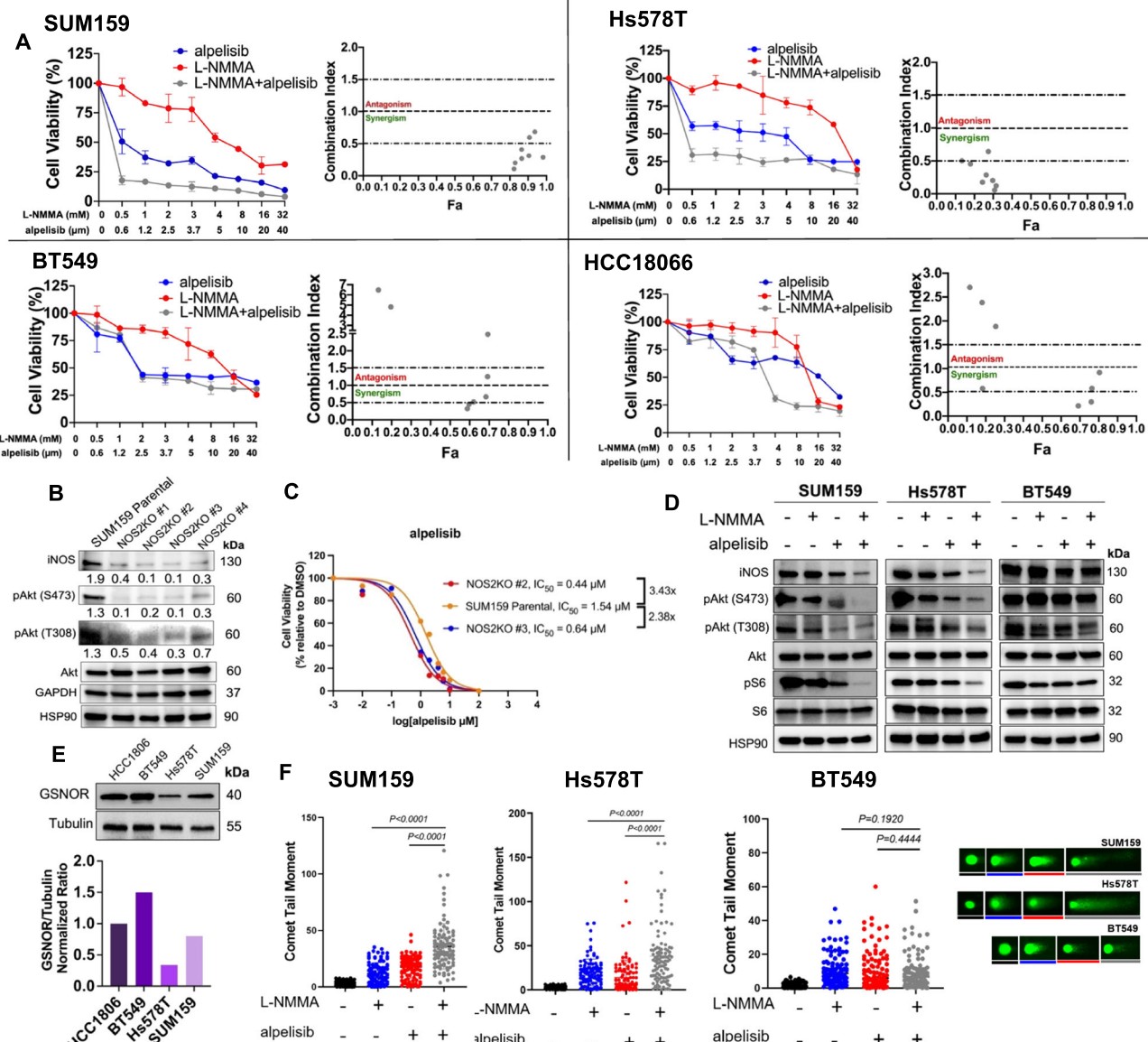

**Fig. 3 | Pan-NOS inhibitor L-NMMA acts synergistically with isoform α-specific PI3K inhibitor alpelisib in MpBC cell lines with *PIK3CA/PIK3R1* mutations.**
**A** Four MpBC cell lines were treated with dimethyl sulfoxide (DMSO), or increasing concentrations of L-NMMA, alpelisib, or combination for 72 hours. Cell growth was evaluated using Sulforhodamine B (SRB) assay. Sensitivity of MpBC cell lines to L-NMMA alone, alpelisib alone, or L-NMMA combined with alpelisib was compared to vehicle control treated MpBC cells. Cell viability (left) and the combination index (right) are shown for each of these four cell lines and determined by CalcuSyn software. Fa, fraction affected. Bars and error bars represent mean ± SD of three biological replicates. **B** Protein levels of iNOS, phospho-Akt (Ser473/Thr308), total Akt, and GAPDH in SUM159 control and different *NOS2* knockout (NOS2KO) clones. NOS2KO clones were developed using iNOS Double Nickase CRISPR plasmids. Blots shown are representative images of *n* = 2 biological replicates. **C** Cell Glo Titer Cell Viability Assay results of SUM159 control and NOS2KO clones treated with alpelisib

at varied concentrations for 72 hours. IC$_{50}$ values were determined by GraphPad Prism software. **D** Immunoblotting of iNOS and PI3K signaling markers in SUM159 (*PIK3CA* mutated), Hs578T (*PIK3R1* mutated), BT549 (PTEN-deleted) cell lines treated for 24 hours with DMSO control, 4 mM L-NMMA, 5 μM alpelisib, and L-NMMA combined with alpelisib. Blots shown are representative images of *n* = 3 biological replicates. Densitometry quantification values were determined using ImageLab software (Biorad) and found in Supplementary Fig. 3. **E** Immunoblotting of S-nitrosoglutathione reductase (GSNOR) and tubulin loading control of MpBC cell lines HCC1806, BT549, Hs578T, and SUM159 with densitometry analysis indicated in a bar graph. The blots shown are representative images of *n* = 2 biological replicates. **F** Extent of DNA damage, quantified by the comet tail moment in the neutral comet assay. Statistical analysis by two-sided Student's *t* test. *n* = 3 biological replicates per condition, 40 comets counted per biological replicate.

cells, NO may increase the transcriptional activity of *JUN* by enhancing the function of JNK through S-nitrosylation. In response to this proposed mechanistic model, we found that NOS2KO cells had a corresponding decrease in JNK-specific phosphorylation of c-Jun at Ser63/Ser73 sites with no associated change in expression of total c-Jun (Fig. 5N) and a decrease in S-nitrosylation of JNK (SNO-JNK) (Fig. 5O). These findings were also confirmed in BT549 cells that underwent CRISPR Cas9-mediated targeting of *NOS2* (Supp. Fig. 5J–L). We also saw evidence of EMT reversal, reduced activation of c-JUN via

phosphorylation, and reduced protein expression of EMT effectors TGFβ and LCN2 in Hs578T cells treated with siRNA targeting NOS2 (Supp. Fig. 5M–O).

## L-NMMA and alpelisib combined with taxane chemotherapy effectively targets breast CSCs
EMT programming can induce the acquisition of breast cancer stem cell (BCSC) properties and MpBC tumors are highly enriched with chemoresistant BCSC populations[32]. Driven by our RNA-seq analysis

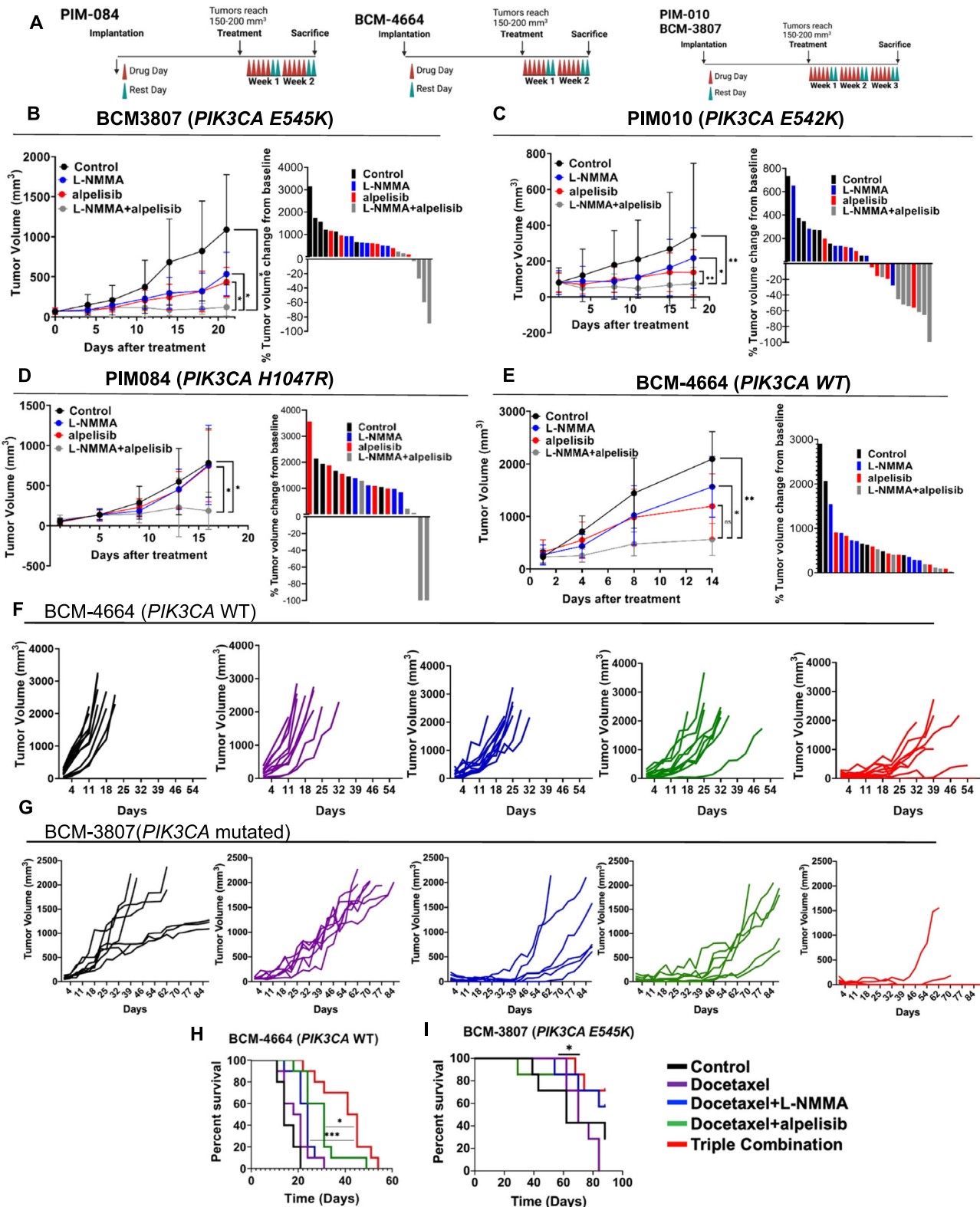

showing downregulation of genes involved in CSC maintenance in SUM159 NOS2KO cells and increased chemosensitivity of MpBC tumors when treated with NOS and PI3K inhibitors (Figs. 4 and 5), we determined whether a combination of NOS/PI3K inactivation with taxane chemotherapy was effective at targeting BCSCs.

For this, we stratified mice bearing BCM-3807 PDX tumors among eight treatment groups: vehicle control, single-agent therapy (L-NMMA, alpelisib, docetaxel), dual-agent therapy (docetaxel+L-NMMA,

docetaxel+alpelisib), and triple combination (docetaxel+L-NMMA +alpelisib). Following a 14-treatment period, we conducted flow-cytometric analysis for BCSC markers, mammosphere-formation efficiency (MSFE) assays, and limiting dilution assays (Fig. 6A).

Primary and secondary MSFE was significantly decreased in tumors with triple combination therapy compared with vehicle treatment (Fig. 6B, C). Triple combination therapy significantly reduced the formation of secondary mammospheres compared to double

**Fig. 4 | NOS inhibition augments PI3K inhibitor and taxane treatment in vivo.**
**A** Schematics representing the MpBC PDX (BCM-3807, BCM-4664, PIM-010, and PIM-084) experimental design. PDXs derived from human MpBCs were transplanted into cleared mammary fat-pad of female NSG mice. When tumors reached 150–200 mm³, mice were randomized to receive vehicle control, NOS inhibition therapy (L-NMMA [400 mg/kg oral gavage on day 1, 200 mg/kg oral gavage on days 2–5] + amlodipine [10 mg/kg intraperitoneal injection on days 1–5]), PI3K inhibitor alpelisib (35 mg/kg oral gavage on days 1–5), or the combination of both therapies as indicated. Caliper measurements were taken twice a week. Days in which mice were treated with therapies are indicated in red and rest days are indicated in green. **B**–**E** Mean tumor volume and corresponding waterfall plots demonstrating maximal treatment response to single-agent or combination therapy in four MpBC PDX models ([BCM-3807, n = 6], [PIM-010, n = 7], [PIM-084, n = 5], and [BCM-4664, n = 6]). Average tumor volume [0.5 × (mm long dimension) × (mm short dimension)²] and data points are mean tumor volume ± SEM. Statistical analysis for **b**–**e** by two-sided Student's t test (*p ≤ 0.05, **p ≤ 0.01). Each bar in waterfall is derived from the maximal response of a single tumor-bearing mouse to therapy. Lines and bars in the plots indicated in black represent vehicle control, blue represent L-NMMA single-agent therapy, red represent alpelisib single-agent

therapy, and gray indicate dual-agent therapy. P values: BCM-3807 (control vs L-NMMA+alpelisib [p = 0.0211], L-NMMA vs L-NMMA+alpelisib [p = 0.0398], alpelisib vs L-NMMA+alpelisib [p = 0.0456]), PIM-010 (control vs L-NMMA+alpelisib [p = 0.006], L-NMMA vs L-NMMA+alpelisib [p = 0.0231], alpelisib vs L-NMMA +alpelisib [p = 0.0079]), PIM-084 (control vs L-NMMA+alpelisib [p = 0.0231], alpelisib vs L-NMMA+alpelisib [p = 0.016]), BCM-4664 (control vs L-NMMA+alpelisib [p = 0.0052], L-NMMA vs L-NMMA+alpelisib [p = 0.0254], alpelisib vs L-NMMA +alpelisib [p = 0.1275]). **F**, **G** Tumor volumes of BCM-4664 (**F**) and BCM-3807 (**G**) tumors treated with vehicle control (black), docetaxel (purple), or combination therapy (docetaxel + NOS inhibition therapy [blue], docetaxel + alpelisib [green], and docetaxel + NOS inhibition therapy + alpelisib [red]). When tumors reached 150–200 mm³, they were randomized into the respective treatment arms. Each graph line represents a replicate/treatment arm. **H**, **I** Kaplan–Meier survival curves of model BCM-4664 (**H**) and BCM-3807 (**I**) treated with vehicle control, docetaxel, or combination therapy (dual/triple combination). An event was scored when a tumor reached 1200 mm³ or from death. Statistical analysis using Log-rank (Mantel–Cox) test. P values: BCM-4664 (docetaxel+alpelisib vs triple combination [p = 0.0432], docetaxel+L-NMMA vs triple combination [p = 0.0007], BCM-3807 (docetaxel vs. triple combination [p = 0.0192].

combination therapy (Fig. 6C). Triple combination treatment also reduced the CD44+/CD24− subpopulation of cells (BCSC marker) and reduced tumor cells capable of tumor initiation, as shown by limiting dilution assay. (Fig. 6D, E, Supp. Fig. 6A). To further evaluate whether triple combination therapy was effective at targeting cells with tumor-initiating potential that may be resident post-docetaxel therapy[6,33,34], we transplanted at limiting dilutions cells from BCM-3807 tumors treated for 14 days with either docetaxel or triple combination therapy, into mammary fat pads of SCID/Beige mice. After 12 weeks, we found that secondary transplantation of cells derived from triple combination therapy resulted in a 17-fold loss of tumor-initiating ability compared to docetaxel therapy only (Fig. 6F).

We performed flow-cytometric analysis of BCSC marker aldehyde dehydrogenase (ALDH1) in cells from responsive BCM-3807 tumors following single-agent and dual-agent treatment. ALDH1+ cells were significantly decreased in tumors after L-NMMA+alpelisib compared to vehicle control (p = 0.0006), L-NMMA (p = 0.036), and alpelisib (p = 0.047) (Fig. 6G, H). We obtained comparable results by immunofluorescence analysis of ALDH1 in responsive PDX BCM-3807 and PIM-010 tumors following single-agent and dual-agent treatment. ALDH1 expression was significantly decreased in dual-agent treated BCM-3807 (p < 0.0001) and PIM-010 (p < 0.0001) tumors relative to vehicle control-treated tumors (Supp. Fig. 6B–E). Overall, our findings suggest that combined NOS and PI3K inhibition is capable of targeting chemoresistant and tumor-initiating population of breast cancer cells within MpBC tumors, leading to their increased chemosensitivity.

### NOS inhibition reverses EMT in MpBC human tumor biopsies

To confirm our in vitro and in vivo findings revealing that NOS inhibition reverses EMT and enhances tumor differentiation, we performed immunofluorescence analysis and evaluated the expression of epithelial marker E-cadherin, ZEB1, and ALDH1 in responder and non-responder tumor biopsies at BL and EOT from the L-NMMA+taxane clinical trial discussed in Fig. 1. We also included corresponding hematoxylin and eosin staining of BL and EOT biopsies of representative non-responder and responder tissues (Fig. 7A, B) in Supplementary Fig. 7. Zeb1 showed a statistically significant decrease in all responders (p < 0.05) with no significant change in non-responders (Fig. 7A–C). Similarly, the increase in E-cadherin from BL to EOT was statistically significant in all responders (p < 0.05), whereas it had no significant change in the non-responders (Fig. 7A, B, D). ALDH1, a marker of mammary stem cell and tumor initiation, showed a statistically significant decrease in two out of the three responders, from BL to EOT (p < 0.05), but no significant change in the non-responders. All non-responders had an elevated BL level of ALDH1 as compared to

responders (Fig. 7A, B, E). Collectively, our findings suggest that iNOS elicits its oncogenic activity in MpBC by acting on a potent activator of EMT and tumor dedifferentiation (Fig. 7F), providing an additional line of evidence of NOS as a highly valuable molecular target in clinical management of MpBC. In our study, we found that in a quasi-mesenchymal/mesenchymal state, iNOS from MpBC cells can produce NO, leading to the attachment of a S-nitrosyl group to a reactive thiol group in JNK, forming a S-nitrosothiol (SNO). SNO-JNK activates the kinase, leading to phosphorylation of AP-1 proteins, such as c-Jun transcription factor, resulting in enhanced expression of EMT mediators *TGFB1* and *LCN2*. TGFβ can activate PI3K/Akt signaling by stimulating Type I and II serine/threonine kinase receptor complex (TβRI/TβRII), causing TβRI to associate with p85, mediating Akt activation[35]. NO can also activate pro-survival signaling pathways, such as PI3K, that influence activation of JNK in a mechanism independent of S-nitrosylation. Under NOS inhibition, reduced activation of PI3K signaling and decreased formation of SNO-JNK result in less activation of c-Jun, and decreased mRNA expression of *TGFB1* and *LCN2*. The reduced expression of EMT mediators results in MpBC cells differentiating back to an epithelial-like state, leading to increased susceptibility to conventional taxane chemotherapy (Fig. 7G).

### Discussion

Despite the use of conventional treatment with surgery and chemo/radiotherapies to treat MpBC, clinical outcomes remain poor[26]. The aggressiveness of MpBC has been attributed to the enrichment of EMT/CSC features, triggered by the activation of oncogenic pathways, including iNOS and PI3K signaling[5,36]. Our findings demonstrated that human MpBC tumors show enhanced co-activation of iNOS and PI3K signaling, and we identified a strong association between increased expression of *NOS2* and a worse MFS in patients with TNBC and MpBC. This initial analysis of human tumors enabled us to propose the dual targeting of the hyperactivated NOS and PI3K pathways as a rational therapeutic strategy to treat chemoresistant MpBC. We found that NOS inhibition augmented the efficacy of PI3K inhibitor alpelisib in MpBC in vitro and in vivo models. Combined NOS and PI3K inhibition further sensitized MpBC tumors to chemotherapy and improved OS regardless of *PIK3CA* mutation status. In MpBC cell line studies, we found that combined PI3K and NOS inhibition had an antagonist effect in BT549 (*PTEN* mutated) and HCC1806 (*PIK3CA/PTEN* wild-type) cell lines. BT549 likely had a poor response to combined NOS and alpha-isoform-specific PI3K inhibition due to its lack of PTEN. This deficiency may have led these cells to be more dependent on the p110β PI3K isoform[37]. Furthermore, the absence of PTEN suggests that BT549 cells may rely less on NO for oncogenic signaling, as PTEN's post-

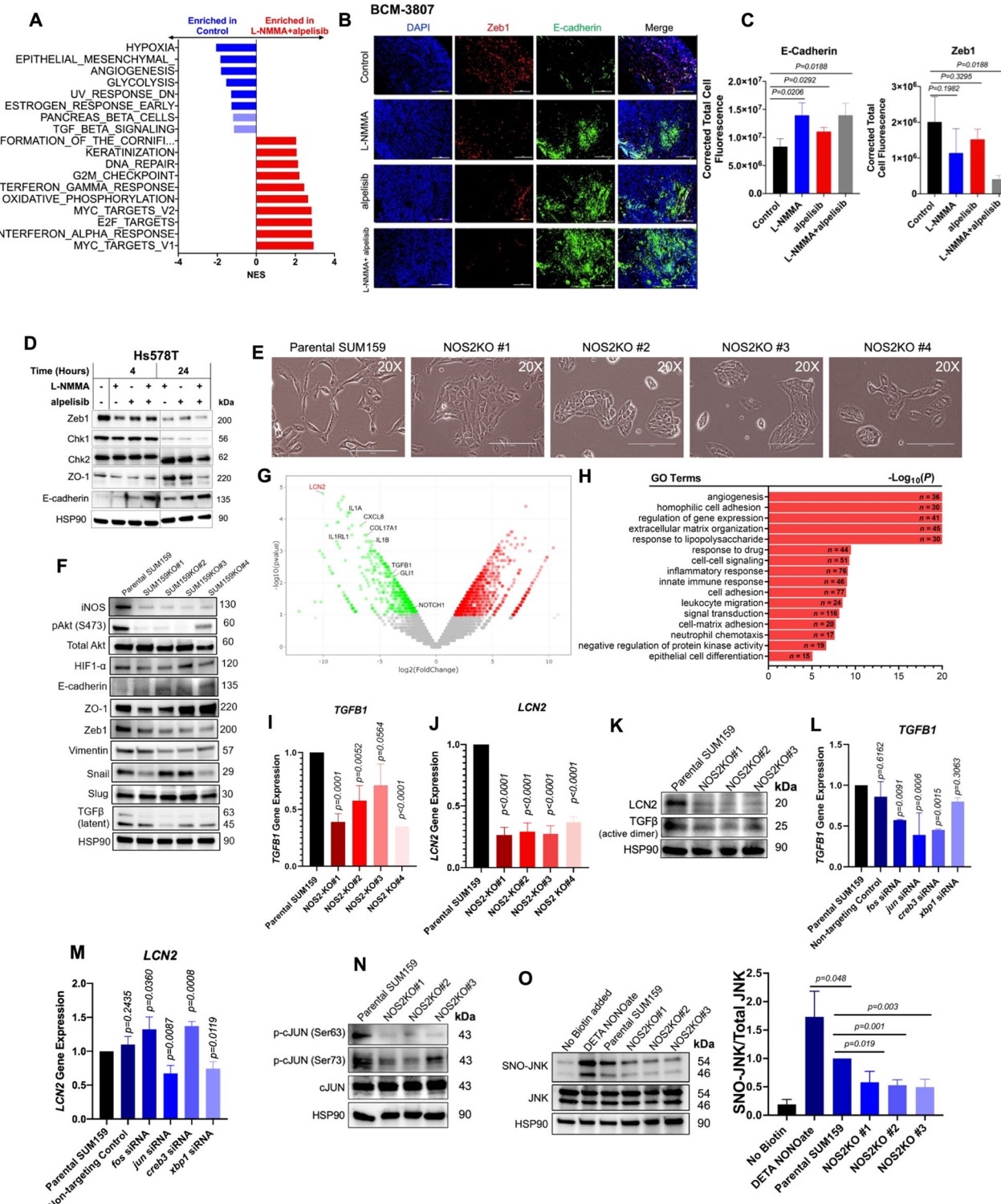

translational inactivation via S-nitrosylation would not be possible[17]. HCC1806 had a poor response to combination therapy, likely due to its low activation of iNOS and PI3K signaling, as exhibited in Fig. 2C as well as its relatively higher expression of GSNOR seen in Fig. 3F; however, the relationship between response to iNOS inhibitors, GSNOR expression, and S-nitrosylation is still unclear[38].

According to our mechanistic studies, NOS inhibition overcomes resistance and augments the efficacy of alpelisib and taxane therapy by primarily inhibiting EMT and decreasing stemness, leading to enhanced tumor cell differentiation. The suppression of EMT in MpBC appears to contribute to the synergistic cytotoxicity and DNA damage elicited by dual NOS and PI3K inhibition in MpBC tumors. It may explain the delayed growth of MpBC tumors in mice treated with L-NMMA and taxane-based chemotherapy, the benefit of MpBC patients who received the same type of therapy, and the significant association of low NOS expression with longer MFS that we observed in the cohort with MpBC patients (Fig. 1H).

In MpBC cells, NOS inhibition decreases the production of NO, thus reducing the formation of SNO-JNK. Reduced SNO-JNK substantially impaired phosphorylation of AP-1 transcription factor c-Jun and decreased the expression of EMT mediators, TGFβ, and lipocalin, resulting in tumor cell epithelization. Furthermore, combining

**Fig. 5 | NOS inhibition induces epithelial-to-mesenchymal transition reversal in MpBC. A** Top GSEA pathways by normalized enrichment score (NES) from the Hallmark and Reactome collections, enriched in control (blue) and L-NMMA +alpelisib (red) treated PDX tumors. Light blue bars indicate non-significant pathways. **B** Representative immunofluorescence images of BCM-3807 tumors evaluated for **C** E-cadherin and Zeb1 protein expression. $n = 3$ biological replicates per condition. Five images at ×10 magnification per biological replicate covering the complete tissue bed were utilized for analysis. Scale bars represent 200 μM. Black, blue, red, and gray bars represent vehicle control, L-NMMA, alpelisib, and L-NMMA +alpelisib, respectively. **D** Immunoblotting of EMT markers in Hs578T (*PIK3R1* mutated) cell lines treated with DMSO control, L-NMMA, alpelisib, and L-NMMA +alpelisib for 4–24 hours. **E** Morphology of SUM159 control cells and NOS2KO clone cells. ×20 magnification and scale bars represent 200 μM. **G** Volcano Plot representing global transcriptional changes comparing SUM159 control cells and NOS2KO clone cells. Each data point represents a gene regulated by AP-1

transcription factor family. Differentially expressed genes ($p < 0.05$) with a log2 fold change >1 are upregulated genes (red dots), and less than −1 are downregulated genes (green dots). Statistical analysis was performed using the Wald test. **H** Significantly differentially expressed genes clustered by their gene ontology (GO) with an adjusted *P* value < 0.05, tested using two-sided Fisher exact test (GeneSCF v1.1-p2). mRNA expression of *TGFB1* **I** and *LCN2* **J** from Parental SUM159 cells and NOS2KO clone cells and *TGFB1* **L** and *LCN2* **M** in SUM159 treated with non-targeting control siRNA and siRNAs specific to *XBP1, CREB3, FOS,* and *JUN*. Immunoblots of **F** EMT and iNOS-associated proteins, **K** LCN2, TGFβ [active form], **N** phospho-c-Jun (Ser63/Ser73), **O** S-nitrosylation of JNK (SNO-JNK), and HSP90 loading control in Parental SUM159 cells and NOS2KO clone cells. For **C, I, J, K–O,** Statistical analysis by two-sided Student's *t* test. Bars and error bars represent the mean ± SD of three biological replicates. For all Blots, images shown are representative of $n = 3$ biological replicates, and graph represents SNO-JNK/JNK protein expression ratios from $n = 3$ biological replicates.

L-NMMA and alpelisib with taxane was superior to taxane alone at targeting chemoresistant CSC populations, resulting in significantly decreased overall tumor burden and improved OS. Additionally, combined targeting of NOS and PI3K reduced cell proliferation, thereby providing a two-pronged attack by reducing invasion through inhibiting EMT and decreasing the proliferation of MpBC tumor cells that may colonize metastatic sites. Lastly, in a phase 1b/2 clinical trial assessing the efficacy of L-NMMA combined with taxane in treating patients with chemorefractory, LABC, or metastatic TNBC[13], we evaluated tumor biopsies of responder/non-responder MpBC tumors. In responder tumor biopsies at EOT, there was an associated enhancement in tissue differentiation/epithelization, along with a decreased expression of CSC marker ALDH1 and iNOS. The sample size of MpBC human tumor biopsies that we evaluated in our study was small ($n = 13$); however, we note that MpBC is a rare and highly aggressive pathology[39]. Although our investigation is confined by the quantity of MpBC PDXs we studied, we observed that those PDXs exhibiting favorable responses to combination therapy primarily exhibited heightened protein expression of iNOS and pAkt, in conjunction with the presence of *RPL39/PIK3CA* mutations. Additionally, they were characterized as matrix-producing metaplastic (PIM-010 and PIM-084) and squamous metaplastic (BCM-3807) carcinomas. Using in vitro/ in vivo approaches and validating with human MpBC tissue, we discovered that inhibition of tumor-intrinsic NOS can overcome chemoresistance via induced tumor differentiation.

In other disease models, it has been reported that iNOS-derived NO can enhance the expression of genes associated with fibrosis through S-nitrosylation-mediated activation of JNK[31,40]. An activated form of SNO-JNK phosphorylates c-Jun, resulting in transcriptional activation of AP-1 downstream genes. Considering the implication that AP-1 regulates the expression of EMT mediators TGFβ and LCN2 in organ fibrosis[41,42], we suspected a similar mechanism for MpBC cells during NOS inhibition. Our study delineated the relationship between iNOS and the major mediators of EMT, thus proposing a unique mechanism of how tumor differentiation processes can be modulated for therapeutic benefit, in cancers like MpBC. Our initial findings using the TCGA database suggested that there may be enhanced iNOS activation in other cancers, such as endometrial and ovarian cancers. We also found increased co-occurrence of mutations in genes from iNOS and PI3K signaling in multiple cancer types and our findings suggest that NOS is upstream of PI3K signaling proteins. This may explain how their co-activation and enrichment may be synergistic in inducing tumorigenesis, cancer progression, and promoting an aggressive phenotype. We have previously discussed the role of NOS in cancer progression and metastases and as a therapeutic target in multiple tumor types, including melanoma, hepatocellular carcinoma, pancreatic ductal adenocarcinoma, head and neck squamous carcinomas, and glioblastomas[36]. In high-grade ovarian cancers, nearly 70% of all cases have hyperactivation of PI3K signaling pathway, high iNOS

expression is predominant in these tumors and may be associated with resistance to platinum-based chemotherapies and enhanced metastatic capacity mediated by EMT[43–45]. Our study is one of the first to show that co-targeting NOS and PI3K may be a rational therapeutic strategy to treat MpBC. We hypothesize that this therapeutic combination may hold relevance for other solid tumors exhibiting concurrent activation of both pathways. We propose that our suggested mechanism involves NOS inhibition leading to EMT reversal by inhibiting JNK S-nitrosylation and that heightened sensitivity of tumors to PI3K inhibitors and taxane therapy could potentially extend to cancers such as ovarian cancer.

Our findings are supported by previous studies describing the influence of iNOS and NO on activation of PI3K signaling regardless of *PIK3CA* mutation status. We previously discussed how iNOS modulates PI3K/Akt signaling through various S-nitrosylation mechanisms[36]. For example, in melanoma models, iNOS-generated NO can reversibly S-nitrosylate the TSC2 protein, disrupting TSC2/TSC1 dimerization, leading to mTOR activation and increased melanoma cell proliferation[46]. Additionally, iNOS-derived NO can S-nitrosylate PTEN protein, reducing PTEN phosphatase activity and promoting PI3K/Akt signaling[17]. Finally, in breast cancer cells, iNOS-associated Akt activation necessitates TIMP1 protein nitration and TIMP1 and CD63 protein-protein interactions[47].

Tipping the EMT programming scale towards more of an epithelial/ proliferative state and targeting CSCs may enhance tumor sensitivity to targeted therapies and chemo/radiotherapies[48–50]. The tumor microenvironment (TME) is a critical source for factors like hypoxia (HIF-1α, NO), and signaling proteins derived from cells such as cancer-associated fibroblasts and macrophages (TGFβ, LCN2, IL-6, IL-1β, etc.) that influence EMT[50]. A limitation of our study is studying MpBC tumors in immunocompromised mice, narrowing our ability to fully evaluate the impact of NOS and PI3K inhibition on TME and its relationship with tumor cell EMT programming. There have been published reports of validated immunocompetent murine MpBC models that recapitulate features of human disease[51–53], however many of these models are not commercially available for further studies and require significant time in development of tumors with metaplastic features. These genetically engineered murine models (GEMMs) have advanced our understanding of the critical molecular drivers responsible for specific metaplastic phenotypes[54]. They have also been shown to be clinically relevant for examining the relationship between MpBC phenotypes and resistance to various therapies, such as PARP inhibitors[53,55]. For instance, Melchor and colleagues demonstrated that GEMMs with *BRCA2* and *TP53* deletions in luminal ER-negative (lumER^neg) cells produced metaplastic spindle cell tumors in 48.3% of cases. Additionally, GEMMs with *PTEN* deletions in lumER^neg cells generated tumors that were 40% adenomyoepitheliomas (AME), 40% metaplastic adenosquamous carcinomas (ASQC), and 20% had elements of both AME and ASQC[54]. These findings reinforce our study and others that highlight PI3K signaling as a crucial driver of MpBC carcinogenesis.

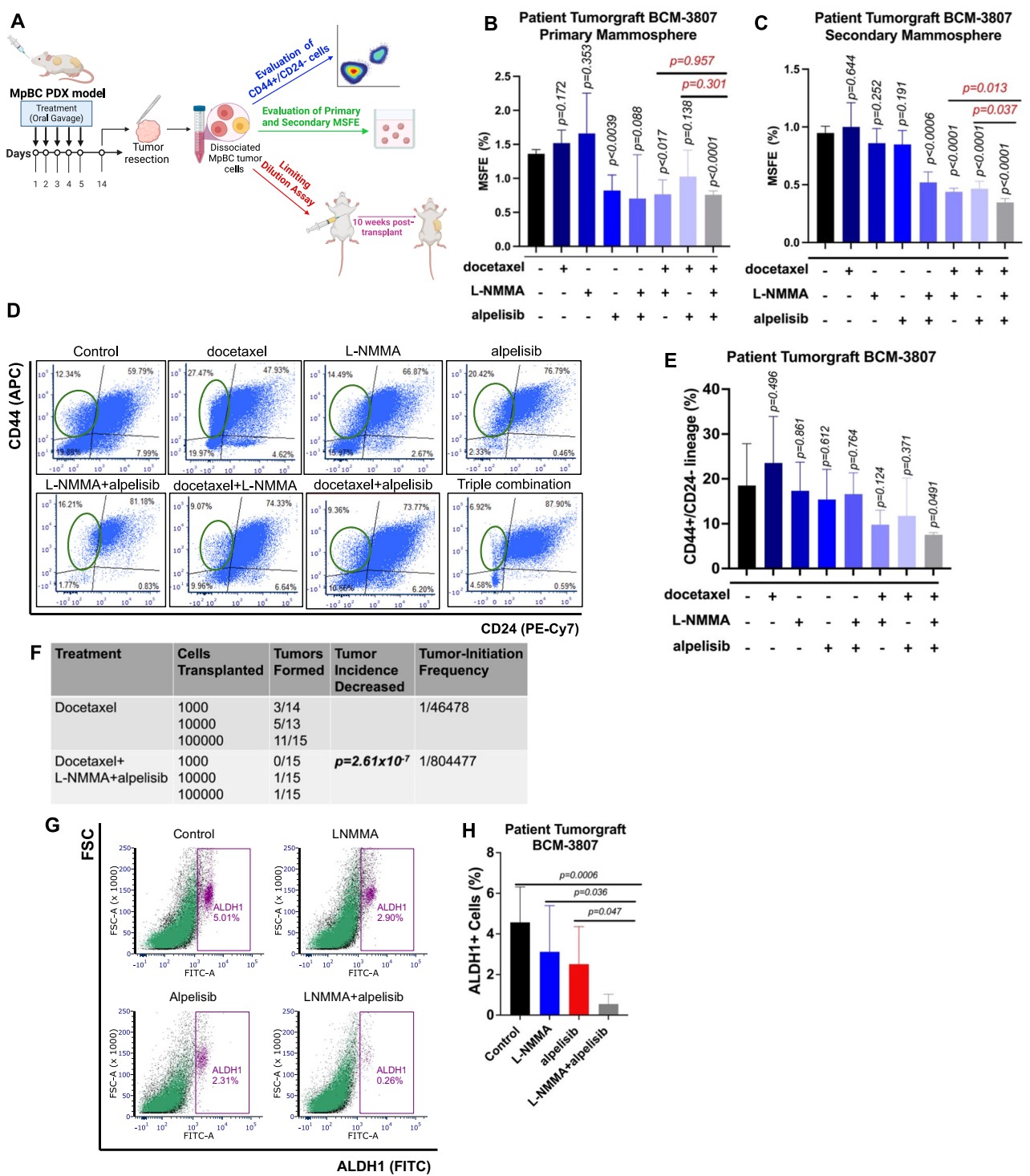

Our study focused on evaluating how targeting tumor-intrinsic iNOS could influence MpBC differentiation and therapy response. Interestingly, our sequencing analysis of MpBC cells lacking *NOS2* revealed a significant decrease in the expression of pro-inflammatory genes (*CXCL8, IL1A, IL1B, IL1RL1*), suggesting that there may be cross-talk between tumor-intrinsic iNOS and TME-derived inflammatory cells that may influence tumor EMT programming. Since MpBC is typically TNBC, analysis of available syngeneic models of this breast cancer subtype may assist in better characterizing aspects of NOS function involving the TME and impacting the clinical effects of NOS-targeted therapies. Collectively, our work nominates combined NOS and PI3K inhibition as a therapeutic strategy for patients with MpBC.

## Methods

### Cells and reagents

Cell lines used in this study were obtained from ATCC, used within three passages since thawing, tested negative for mycoplasma, authenticated by short tandem repeat profiling, and cultured in media containing 10% FBS (GenDepot) and 1% antibiotic-antimycotic reagent (GenDepot). Media used for cell lines is indicated in Supplementary Table 3. L-NMMA was obtained from the cGMP facility at Houston Methodist Research Institute, PI3K inhibitor alpelisib was purchased from Cayman Chemicals, and docetaxel was purchased from Houston Methodist Pharmacy. Cells were treated with inhibitors in serum-free media (1% FBS) for specific time points indicated in the results section.

**Fig. 6 | L-NMMA and alpelisib combined with taxane chemotherapy are effective at targeting breast cancer stem cells. A** Schematic showing experimental outline to test the tumor-initiating and self-renewal capacity of MpBC PDX tumor model BCM-3807 after treatment with targeted therapies with or without taxane chemotherapy. Figure created with Biorender.com. **B** Primary and **C** Secondary MSFE % values of tumors from each indicated treatment group after 14 days of treatment. Tumors were treated in vivo for 14 days, collected, dissociated into single cell suspensions, and subsequently plated under mammosphere conditions (60,000 cells per well in 6-well low-adherent plates in supplemented MammoCult Media). To determine secondary MSFE, primary mammosphere were collected, dissociated, and replated under mammosphere conditions. A two-tailed Student *t* test was conducted to evaluate *p* values comparing each treatment condition to vehicle control. Bars and error bars represent the mean ± SD of four biological replicates/mice per condition. *P* values indicated in red compare triple combination therapy to both double combination therapies. *P* values indicated in black compare indicated treatment arm to vehicle control. **D, E** CD44+/CD24− (BCSC marker)

results (flow cytometry) from tumors after treatment. A two-tailed Student *t* test was conducted to evaluate p values comparing each treatment condition to vehicle control. Bars and error bars represent the mean ± SD of four biological replicates/ mice per condition. **F** Limiting Dilution Assay: BCM-3807 tumors from docetaxel and triple combination (docetaxel+L-NMMA+alpelisib) treated mice (14 days) were dissociated and pooled. A total of 1000, 10,000, and 100,000 cells from each group were transplanted into the mammary gland fat-pad of 4- to 6-week-old mice (*n* = 13–15 mice/group). Tumor incidence was reported at 12 weeks post-transplantation. Stem cell frequency fractions and overall statistical analysis using Chi-squared test comparing differences in stem cell frequencies between treatment groups was evaluated using available LDA software (https://bioinf.wehi.edu.au/software/elda/). **G, H** Flow cytometry analysis of ALDH1+ cells in BCM-3807 tumors after 14 days of treatment with vehicle control, single-agent therapy (L-NMMA or alpelisib), and dual-agent therapy. Statistical analysis by two-sided Student's *t* test. Bars and error bars represent the mean ± SD of five biological replicates/mice per condition.

For the CRISPR knockout of *NOS2*, cells were transfected with the *NOS2* Double Nickase Plasmid (Santa Cruz Biotechnology). After puromycin selection, resistant clones were selected by immunoblotting for iNOS expression and nitrite/nitrate colorimetric assay (Sigma). For siRNA knockdown, cells were transfected in a 6-well-plate with ON-TARGETplus SMARTpool human CREB3, XBP1, FOS, and JUN siRNA pools or non-targeting siRNA pools as control (Horizon Discovery, Dharmacon) using Lipofectamine RNAiMax (ThermoFisher). For TGFβ rescue experiments, SUM159 Parental and NOS2KO cells were treated with TGFβ (5 ng/ml; Cell Signaling) for 24 hours.

For qPCR analysis, RNA was isolated from breast cancer cell lines and PDX models using Qiagen RNeasy Kit according to manufacturer's instructions. cDNA synthesis was performed using Superscript IV VILO Master Mix (ThermoFisher), and qPCR was performed using Power-SYBR Green PCR Master Mix (Applied Biosystems). The ddPCR experiments were performed using a QX200 ddPCR system (BioRad) as described previously[8] and analyzed with QuantaSoft software. Information about cell lines, primers, and antibodies are provided in Supplementary Tables 3–6.

### In vitro assays
For cell viability assay, treated MpBC cells were analyzed with Sulforhodamine B assay following manufacturer's instructions (Abcam) and Cell Titer Glo assay (Promega). For Caspase 3/7 assay, MpBC cells were treated with inhibitors at physiologically relevant concentrations for 72 hours (L-NMMA at 4 mM, alpelisib at 5 μM, or combination of two drugs), and then incubated with the caspase 3/7 reagent, following manufacturer's instructions (Promega). MpBC cell lines (SUM159, Hs578T, HCC1806, and BT549) were used for drug combination assay. L-NMMA and alpelisib were each arrayed in 96-well plates and serially diluted two-fold, yielding a concentration equal to $IC_{50}$ of each drug in MpBC cell lines. Cells were seeded in 96-well plates at a concentration of $3.0 × 10^4$ cells/mL in 100 μL of medium per well, and then treated with a single drug or with a combination of L-NMMA and alpelisib for another 72 hours. When testing for synergy of the drug combination, L-NMMA and alpelisib were plated serially onto MpBC cells with constant ratio concentrations in 96-well plates with Eppendorf Xplorer plus Electronic Single Channel Pipette. CI and fraction affected (Fa) values were calculated using CalcuSyn software. GraphPad Prism 6 software was used to plot dose-response and caspase 3/7 activation plots and determine $IC_{50}$ concentration. For Anchorage-Dependent Colony Formation Assay, MpBC cells were plated and treated with inhibitors accordingly. After 72 hours of inhibitor treatment, cells were washed with ice-cold PBS, fixed with ice-cold 100% methanol, stained with 0.5% crystal violet solution, washed, and dried well images were captured. Neutral comet assays were performed using the CometAssay Kit (Trevigen), as per the manufacturer's instructions and as we previously reported[56]. Comet tail area was measured using CaspLab

software, and calculations were averaged from three independent experiments.

### Rad51 and γH2AX foci immunofluorescence
Immunofluorescence was performed following standard protocols[57]. MpBC cells were treated for 24 hours with DMSO control, L-NMMA, alpelisib, or combination (L-NMMA+alpelisib) on chambered cell culture slides (Corning). Cells were fixed, blocked, incubated with primary/secondary antibodies, and mounted coverslips using Antifade Mounting Medium with DAPI (Vector Laboratories). Images were acquired with Nikon A1R confocal imaging system with NIS Elements software (Nikon). Rad51 and γH2AX foci were quantified in 100 cells per replicate using ImageJ software.

### Nucleotide quantification
MpBC cells were seeded at a density of $2×10^6$ cells and treated with DMSO, L-NMMA, alpelisib, or L-NMMA+alpelisib for 8 hours. After treatment, cells were prepared for dNTP quantification with method described previously[58].

### Biotin-switch assay
A modified biotin-switch assay was performed as described previously[59]. SNO-JNK was detected using an S-nitrosylated protein detection kit (Cayman Chemicals, #10006518) according to manufacturer's instructions and all steps were carried out in the dark. Biotinylated proteins were then purified with NeutrAvidin Plus Ultralink Resin (ThermoFisher), separated by SDS-PAGE, and immunoblotting to detect JNK expression.

### RNA-sequencing and gene set enrichment analysis (GSEA)
RNA samples were extracted with Qiagen RNAeasy Kit following manufacturer's instructions. RNA was sent out to commercial companies, Novogene and Genewiz for library preparation, high throughput sequencing using Illumina sequencers, and analysis (gene ontology/GSEA and differential gene expression analysis, https://www.gsea-msigdb.org/gsea/index.jsp). ARTEMIS dataset (MD Anderson Cancer Center) was analyzed using R Survminer package for survival analysis. Kaplan-Meier method was used to draw survival curves, and the log-rank test was performed to evaluate survival differences.

### Immunoblotting
Whole cell lysates were made in 1× RIPA Buffer (Sigma) containing Protease and Phosphatase Inhibitor Cocktail Solutions (GenDepot). Samples were boiled in 4× Laemmli Sample Buffer (Biorad) and subjected to SDS-PAGE electrophoresis (BioRad). Proteins were transferred onto activated PVDF membranes (BioRad). Membranes were incubated overnight at 4 °C with primary antibodies (see Supplementary Table 5) and HRP-conjugated secondary antibodies for 2 hours, and proteins were visualized using ChemiDoc Imaging System (BioRad).

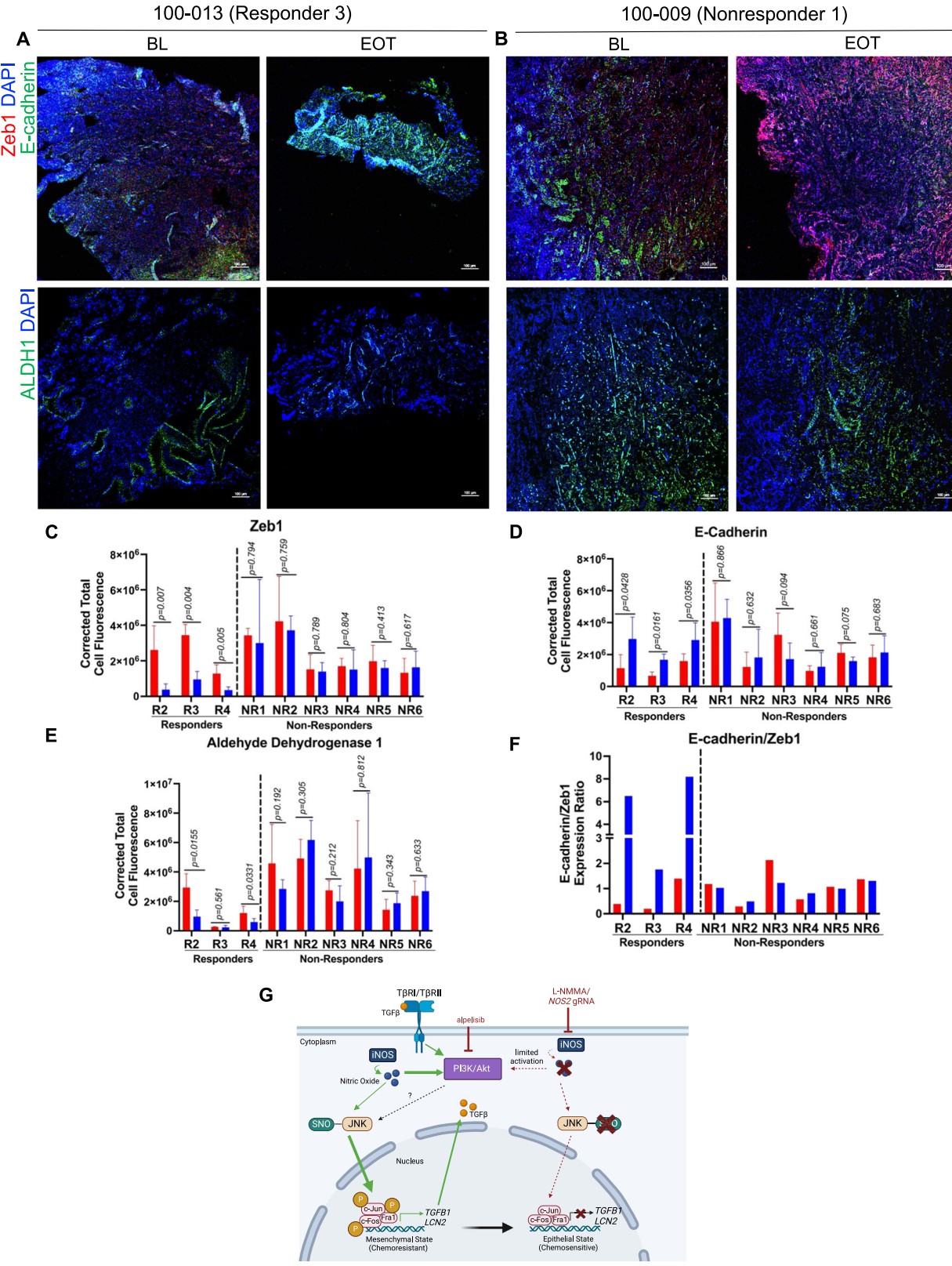

## IHC/immunofluorescence

IHC was performed as described previously, and antibody details are found in Supplementary Table 6[13]. Briefly, tissues of PDX tumors (treated with vehicle control or inhibitors [L-NMMA, alpelisib, L-NMMA +alpelisib]), as well as BL and EOT tissue from nine available phase 2 dose patients (three responders and six non-responders), were evaluated by IHC for iNOS, phospho-Akt (Ser473), and PTEN expression.

FFPE tissues were prepared and underwent xylene deparaffinization, ethanol rehydration, antigen retrieval, fixation with hydrogen peroxide, blocking, antibody incubation, chromogen reaction, and counterstained with hematoxylin, slides were mounted and imaged under bright field microscopy. iNOS staining was analyzed by H-score analysis using ImageJ plugin IHC Profiler, as described previously[60]. For immunofluorescence analysis, the same FFPE tissues underwent

**Fig. 7 | EMT reversal in MpBC human tumor biopsies after L-NMMA+taxane therapy. A** Representative immunofluorescence images of tumor biopsy for patient 100-013 (responder) and **B** patient 100-009 (non-responder) stained for Zeb1 (red), E-cadherin (green), ALDH1 (green), and DAPI (blue). Images magnification ×10; scale bars represent 100 μM. Corrected total cell fluorescence analysis quantified using ImageJ software for **C** Zeb1, **D** E-cadherin, **E** ALDH1 in patient tumor biopsies at baseline (BL) and end-of-treatment (EOT). Immunofluorescence analysis was obtained from 5 images per slide captured from tissues to cover the entire tissue bed. Statistical analysis by two-sided Student's $t$ test. Bars and error bars represent ±SD **F** Ratio of E-cadherin/Zeb expression (corrected total cell fluorescence) in patient tumor biopsies at BL and EOT. Statistical analysis was performed by Student's $t$ test. Red and blue bars indicate data for BL and EOT corrected total cell fluorescence, respectively. R2-R4 indicate responder patients, and NR1-NR6 indicate non-responder patients. **G** Schematic represents a potential mechanism of action of NOS inhibition inducing EMT reversal. iNOS inducible nitric oxide synthase; EMT epithelial-to-mesenchymal transition; JNK c-Jun N-terminal kinase; PI3K phosphoinositide 3-kinase; TGFB1 transforming growth factor beta 1; LCN2 lipocalin AP-1 activator protein 1. Figure created with Biorender.com.

deparaffinization and antigen retrieval as previously described, with no hydrogen peroxide fixation or incubation with primary/secondary antibodies. Slides were mounted with Antifade Mounting Medium with DAPI (Vector Laboratories) and sealed with coverslips.

## In vivo experiments

All animal procedures were approved by the Houston Methodist Hospital Research Institute Animal Care and Review Office (IACUC number IS00007220). The maximal tumor volume permitted in our study is 2000 mm³ as calculated from caliper tumor measurements. All procedures were performed in accordance with the protocol and none of the tumor volumes in our study exceeded 2000 mm³ at the time point before the last time point when the studies were terminated. Some tumors exceeded this limit at the last time point, and these animals were immediately euthanized. In vivo experiments were conducted in four human MpBC PDX models (BCM-4664, BCM-3807, PIM-010, and PIM-084). PDXs were implanted into cleared mammary fat-pad of SCID/Beige mice (Envigo). When tumors reached an average volume between 150 and 250 mm³, mice were randomized into treatment arms. Mouse weights were record and tumor volume was measured and calculated [$0.5 \times$ (long dimension) × (short dimension)²] twice weekly. Tumor volume change was calculated by dividing change in tumor volume at last measurement by initial tumor volume. Regiment treatment design followed either two or three, 2-week cycles of docetaxel (20 mg/kg intraperitoneal on day 1), NOS inhibition therapy (L-NMMA [400 mg/kg oral gavage on day 2 and 9, 200 mg/kg oral gavage on days 3–6 and 10–13] + amlodipine [10 mg/kg intraperitoneal injection on days 2–6 and 9–13]), PI3K inhibitor alpelisib (35 mg/kg oral gavage on days 2–6 and 9–13), or the combination of therapies as indicated. Caliper measurements were taken twice a week.

## Breast CSC assays

Mice implanted with BCM-3807 PDX tumors of 150–250 mm³ were randomized to receive one cycle (two weeks) of docetaxel, docetaxel +L-NMMA or alpelisib, and triple combination. Dissociated tumors were transplanted with Matrigel (1:1) into the fat pad of SCID/Beige mice, as described previously, for limiting dilution assays[61]. For limiting dilution assay experiment in Supplemental Fig. 6, we transplanted 30,000 and 50,0000 PDX-3807 tumor cells into the fat-pad of SCID/Beige mice ($n = 5$–10 per group) from tumors treated for two weeks with docetaxel, docetaxel+L-NMMA or alpelisib, and triple combination. For the limiting dilution assay experiment in Fig. 6, BCM-3807 tumors from docetaxel and triple combination (docetaxel+L-NMMA +alpelisib) treated mice (14 days) were dissociated and pooled. A total of 1000, 10,000, and 100,000 cells from each group were transplanted into the mammary gland fat-pad of 4- to 6-week-old mice ($n = 13$–15 mice/group). Tumor incidence for both experiments was reported at 12 weeks post-transplantation. Stem cell frequency fractions and overall statistical analysis comparing differences in stem cell frequencies between treatment groups were evaluated using available LDA software (https://bioinf.wehi.edu.au/software/elda/).

For flow cytometry, dissociated tumor cells post-treatment (two weeks) were prepared at a concentration of 1 million cells per 0.1 ml in Hank's Balanced Salt Solution (HBSS) supplemented with 2% fetal bovine serum (HBSS+). These cells underwent labeling procedures using fluorophore-conjugated antibodies, including CD44-APC (at a 1:10 dilution), CD24-PECY7 (at a 1:40 dilution), or CD24-FITC (at a 1:10 dilution), or H2kD-PE (at a 1:40 dilution) to eliminate mouse cells. Labeling was conducted either on ice for 15 minutes or at 37 °C for 45 minutes, following the manufacturer's protocol for the Aldefluor kit (StemCell Technologies, B.C., Canada). Additionally, post-Aldefluor or DEAB labeling, H2kD-PE staining was performed on ice for 15 minutes in 0.1 mL HBSS+. Subsequently, cells were washed and treated with propidium iodide at a concentration of 10 μg/mL prior to analysis or sorting using a four-laser FACS AriaII system (BD Biosciences, San Jose, C.A., USA). Debris and doublets were excluded based on side scatter and forward scatter parameters, while propidium iodide staining was utilized to eliminate non-viable cells. Tumor cells negative for H2kD were further characterized for CD44 and CD24 expression or aldehyde dehydrogenase activity. Data analysis was conducted using FACS Diva software (BD Biosciences) and FCS Express (Denovo Software).

For MSFE assays, dissociated tumor cells (60,000) were plated in MammoCult Media (Stem Cells) in low-adherent 6-well and 24-well plates. Mammospheres were imaged and counted using GelCount imaging and software system (Oxford Optronix Ltd.). For secondary MSFE, mammospheres were collected, dissociated with trypsin, filtered, counted, and replated at 60,000 cells/mL MammoCult Media, as described previously[61].

## Statistical analysis

Two-tailed Student $t$ test or Mann–Whitney test, with the mean values and the SD calculated for each group, was performed for comparisons between two groups. One-way ANOVA was performed for multiple group comparisons. Two-way ANOVA was used for all animal experiments. To account for multiple comparisons, Tukey's multiple comparison tests for one-way ANOVA and Bonferroni post tests for two-way ANOVA were performed. Analysis was conducted using Graphpad Prism 5.0 (Graphpad Software Inc.) and Stata V16.1 (StataCorp). In all cases, a two-tailed $P$ values $< 0.05$ were considered statistically significant.

# Data availability

RNA-seq data from SUM159 NOS2KO clones and parental cells are available in the Gene Expression Omnibus (GEO) under accession number GSE234512. RNA-seq data from MpBC PDX tumors have been deposited in the Database of Genotypes and Phenotypes (dbGAP) under accession number phs003814.v1.p1. These data are available for research purposes and under restricted access to protect individual privacy. Researchers holding permanent positions equivalent to tenure-track professors or senior scientists with administrative and oversight responsibilities can apply for access via dbGAP. Requests, which are reviewed by the NCI's Data Access Committee, are generally approved within two days and provide access for up to 12 months. The ARTEMIS data are available from the corresponding author of the Yam et al. (2022) study under a Materials Transfer Agreement with MD Anderson Cancer Center. Data analyses conducted via cBioPortal (https://www.cbioportal.org/) and STRING utilized publicly accessible datasets. Additional data supporting the findings of this study are provided in the Article, Supplementary Information, or Source Data file associated with this publication. Source data are provided with this paper.

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

## Acknowledgements

We thank David Huston and David Threadgill for their helpful suggestions. We thank Shu-Hsia Chen and Yitian Xu from the Houston Methodist Center for Immunotherapy Research for help with flow cytometry sorting and analysis. We thank Matthew Vasquez, Hong Zhao, and Stephen Wong from the Houston Methodist Advanced Cellular and Tissue Microscopy Core for help with confocal and fluorescence imaging capture and analysis. We thank Rajeev Singh and Pam McShane from Houston Methodist Biorepository Core for help with tissue acquisition. We thank Zannatul Ferdous for helping with the manuscript revisions and helpful suggestions. This work was supported, in whole or in part, by NCI grant no. U01 CA268813 (To J. Chang, F. Meric-Bernstam, D. Wink, S. Lockett, S. Wong), R01CA284315, the Breast Cancer Research Foundation (BCRF), CREDO (to J. Chang), R01CA237200, R21CA267386 from NCI (to C. Thomas), R01AI136581 and R01AI162633 from NIAID (to B. Kim), Contract #75N91019D00024 (to S. Lockett), and philanthropic support from Dr. M. Neal and R. Neal.

## Author contributions

Conceptualization was inputted from T.R., A.P., C.T., R.R., S.M., F.M., J.C., C.Y., D.W., S.L., S.W., and J.C.C. Methods were designed by T.R., L.G., C.T., W.Q., J.Z., R.R., and J.C.C. Experiments were conducted by T.R., L.G., C.T., W.Q., J.Z., H.Z., B.M., A.O., Y.C., B.K., J.T., K.M., M.F.C., C.A., N.G., S.K. MpBC PDX models were kindly gifted to us by H.P.W. and S.M. Data analysis was performed by T.R., A.P., L.G. C.T., J.C., H.Z., J.C., and J.C.C. Manuscript was written by T.R. A.P., C.T., and J.C.C. substantially revised the manuscript and approved the submitted version.

## Competing interests

C. Yam reports grants from Conquer Cancer Foundation (Career Development Award supported by Fleur Fairman; the Gianni Bonadonna Breast Cancer Research Fellowship) and other support from MD Anderson Cancer Center (The University of Texas, Houston, TX) during the conduct of the study, as well as other support from Amgen, Merck, Genentech, and GSK outside the submitted work. S. Moulder also reports other support from Eli Lilly and Company and grants from ASCO Career Development Award Mentor outside the submitted work. J. Chang reports grants from the Cancer Prevention and Research Institute of Texas and from NIH during the conduct of the study. J.Chang is the sole inventor on patent application no. 10420838 entitled "Methods for treating cancer using iNOS-inhibitory compositions" held by Houston Methodist Hospital. C. Thomas reports support from NIH during this study. F. Meric-Bernstam reports grants and personal fees from F. Hoffmann-La Roche/Genentech during the conduct of the study, as well as personal fees from AbbVie, Aduro BioTech Inc., Alkermes, AstraZeneca, DebioPharm, eFFECTOR Therapeutics, IBM Watson, Infinity Pharmaceuticals, The Jackson Laboratory, Kolon Life Science, Lengo Therapeutics, OrigiMed, PACT Pharma, Parexel International, Pfizer Inc., Samsung Bioepis, Seattle Genetics Inc., Tallac Therapeutics, Tyra Biosciences, Xencor, Zymeworks, Black Diamond, Biovica, Eisai, Immunomedics, Inflection Biosciences, Karyopharm Therapeutics, Loxo Oncology, Mersana Therapeutics, OnCusp Therapeutics, Puma Biotechnology Inc., Silverback Therapeutics, Spectrum Pharmaceuticals, and Zentalis and grants from Aileron Therapeutics, Inc., AstraZeneca, Bayer Healthcare Pharmaceutical, Calithera Biosciences Inc., Curis Inc., CytomX Therapeutics Inc., Daiichi Sankyo Co. Ltd., Debiopharm International, eFFECTOR Therapeutics, Guardant Health Inc., Klus Pharma, Takeda Pharmaceutical, Novartis, Puma Biotechnology Inc., and Taiho Pharmaceutical Co. outside the submitted work. No disclosures were reported by other authors.
