## [Peer Review File · Nature Communications]

Reviewers' Comments:

Reviewer #1:

Remarks to the Author:

Reddy et al. describe a possible "Achilles heel" of metaplastic breast cancer (MpBC) in the form of nitric oxide synthases (NOS). They could show that NOS inhibition increases the response to phosphoinositide 3-kinase (PI3K) inhibitor alpelisib as well as taxane in vitro and in vivo. The group goes on to show mechanistic evidence of the NOS blockade acting through repressing the JNK/c-Jun pathway. This likely increases chemosensitivity and is responsible for slower and less tumor growth in vivo. Reddy and colleagues investigate this effect in detail in cancer stem-like cells and show a reversal in epithelial-mesenchymal transition (EMT) as well as a decrease in cancer stem cell markers when investigating NOS-repressed MpBC cell lines.

Overall this story makes for a very interesting read with a host of data supporting the authors model. The writing is concise and conclusive and does not over-interpret the data which is great to see these days. The manuscript would benefit greatly from a more thoroughly written methods section, there is some information missing (see below). In addition some experiments seem necessary to support the conclusions which I detailed below. In conclusion

Major revision:

- Figure 1E, add PTEN and pAKT IHC for EOT
- Limiting dilution TP in 6F does not help to make the point. An ELDA analysis requires three not two points to generate a decent output. I suggest 100000, 10000 and 1000 cells for transplants
- 3E: Explain why the NOS Inhibitor L-NMMA has almost the exact same effect than the PI3K Inhibitor alpelisib on iNOS, it is difficult to then argue synergy. Maybe this should be phrased more careful
- 4F/G: it is hard to imagine, that the only difference between 4F and 4G is the PIK3CA mutation. The way I understand these models derived from patients with more than just one mutation. The conclusion that the response (which in itself is not that dramatic) is solely based on PIK3CA seems a little far-fetched.
- 5F the E-cadherin change seems almost not present in the SUM159 KO cells, whereas HS578T changes in E-cadherin seem quite dramatic. So why not add more cell lines to the experiments in Figure 5, that would also add quality and reproducibility to Figure 5G now being derived from a single cell line
-

Minor revision:

- Figure 1F NR6 is unlikely to be non-significant, please correct like stated in the text
- Figure 1H and 1I are misrepresented in the text, please correct
- Line 223 seems to address Figure J on F
- 3B the figure is too small to identify the effect. Rather remove it, put a higher resolution in the supplementary files and leave the quantification in the main files.
- 5F pAKT Thr308 seems to be missing here
- 6B is not a mammosphere assay when 60000 cells can cluster in a plate, good colony forming assays in Methylcellulose or Matrigel or Agar can identify the actual stem cell number in a given sample.
- Why would the single docetaxel/L-NMMA and alpelisib treatment increase the number of CSCs in 6D?
- There is a discrepancy between representative images in 6D versus the evaluation in 6E.
- Reading total ALDH1 fluorescence is not an adequate test, please use the FACS analysis if you wish to show ALDH1 expressing cell changes
- Method section does not adequately describe the xenografts in Figure 4 and Figure 6: cell number? Size of a tumor piece? More detail is required here.

I would support publication of the manuscript given that these couple of issues are adequately addressed.

Reviewer #2:

Remarks to the Author:

Comments to the authors:

In the manuscript "NOS inhibition sensitizes metaplastic breast cancer to alpelisib and taxane chemotherapy by reversing c-JUN-mediated epithelial-to-mesenchymal transition," the authors demonstrate that decreasing cellular levels of nitric oxide through inhibition of iNOS, which is overexpressed in many breast cancer subtypes with poor prognoses, may offer a method of treating disease. Specifically, they find that NOS inhibition reduces deleterious transcriptional changes promoted by JNK and c-Jun, leading to higher differentiation, decreased EMT, reduced cancer stem cell behavior, and enhanced sensitivity to chemotherapy modalities. In all, this work offers incremental improvement to existing therapy and proposes a molecular model through which this new approach may act.

Clarifications:

1. What exactly is the effect of iNOS on PI3K and how does this differ with PI3KCA mutation? Is the kinase S-nitrosylated?
2. GSNOR expression (and SCOR/AKR1A1 if possible) should be assessed in cells and PDX. GSNOR is often downregulated in breast cancers and may help explain the variable response to NOS inhibition.

Additional comments:

1. Attention to various details throughout the paper and consistency in representation should be made. A few examples:
 - The p-value listed in the text for Figure 1I does not match that shown in the figure
 - The text does not mention Figure 1J (it is listed as 1F instead)
 - There is not consistency on hyphenating L-NMMA and c-Jun, etc.
 - Figure 6 text and legend does not seem to point to the correct data for labels of 6G-F (should be H)
 - Figure legends are generally sparse and do not include details such as the colors of the bars and abbreviations used in Figure 7
 - Certain figures that include p-values above bars express the actual p-value, while others simply indicate "p<0.05." An effort should be made to consistently show the true values atop all such bars.
2. Figure 1F: it seems hard to believe that there is no significance between the points in sample NR6
3. Figure 2B: please explain the relevance of correlated mutations in these pathways and how their enrichment in pan-cancer implicates an interplay between NOS and these respective pathways.
4. Figure 2E: since iNOS is graded into buckets for relative expression, could the same be done for pAkt as well to make the correlation easier to visualize?
5. Figure 3C: there is significant leaky expression of iNOS in the iNOS knockout lines. Is the knockout only in a single allele?
6. Figure 3D: are there statistics that can be used to describe the difference between the curves? They are visually quite similar but may hold important differences.
7. Figure 3E: This figure was difficult to interpret and may benefit from some of the following modifications:
 - Removal of the 4-hour data, since much of it does not show the concluded trends that are clear at 24 hours
 - Quantification of Western blots with error bars for replicate experiments. This should be included in general for Western blots.
 - Clarification on certain data: why L-NMMA treatment alone reduces iNOS expression significantly; why SUM cells do not have a reduction in pAkt for Thr (only Ser) but the other cells have

reductions in both; why pAkt increases with L-NMMA alone at 24-hours; why SUM responds the worst with these readouts but seems to exhibit the best response for DSBs in the subsequent Figure 3F

8. Figure 4A: the numbers under the day arrowheads are undefined. These may indicate weeks, but one of the clusters includes 8 arrowheads. Please clarify in figure legends.
9. Please mention the selection of particular cell lines or xenograft models used for certain experiments. For example, Figure 4B-E shows four models with varying degrees of response, while subsequent experiments use different groupings of two models.
10. Figure 5B and other immunofluorescence: labels above the columns would make it easier to read the data, and indicating the merged column.
11. Figure 5G: the significance threshold is listed as $p < 0.1$. Are these values corrected for multiple statistical tests? If not, perhaps a more stringent threshold could be assigned.
12. Figure 5O: this experiment does not have any replicates with error bars and should be done at least three times independently. The SNO protein should be expressed as a densitometry ratio to the total JNK protein, which is first normalized to the HSP90 control, not as a ratio to the HSP90 control. The axis has a typo in the name of the HSP90 protein.
13. Figure 6B/C: the statistical comparisons in these figures are for each group compared to the untreated controls. If the case is to be made that the triple therapy is more beneficial than any of the double therapies, especially the double therapy that does not include L-NMMA, then a comparison between the triple therapy and the double therapies should be included as well. That the triple therapy is also different than the untreated control does not help to build a case for the inclusion of L-NMMA. At least the comparison between the triple therapy and the bar immediately to its left should be shown.
14. Figure 7G: the figure legend and text of the paper describe roles for Akt that are not illustrated in the figure. Inclusion of something for Akt would help summarize the findings.
15. Line 273 mentions a reverse trend (antagonistic) in two cell lines. An explanation of why this unexpected reversal in these cells would be helpful.
16. Line 303 uses the phrase NOSi and PI3Ki, which presumably mean "inhibition," although this is not defined. NOSi may be confusing due to its similarity to iNOS.
17. Lines 308-309, 341: the order mentioned in the text is different than the order mentioned in the figures
18. Discussion: please provide a summary of the scope of this finding within the context of other cancers and other potential targets of iNOS inhibition. One such candidate is ovarian cancer, which may share some key similarities with estrogen and iNOS.

Reviewer #3:

Remarks to the Author:

This is an interesting and exciting study demonstrating that NOS inhibition sensitizes metaplastic breast cancer to alpelisib and taxane chemotherapy and the underlying potential mechanisms, which involve reversal of EMT and stemness. The most important aspect of the study is the synergism of the drug combination that offers potential therapeutic strategy for the highly aggressive breast cancer subtype termed metaplastic carcinoma. Following are several comments that in my view would enhance the study.

1. Metaplastic breast carcinomas have several well-defined pathological subtypes, which are not really investigated in this study. A detailed recent proteogenomic study (ref 3) demonstrates that these subtypes have common as well as distinct proteomic profiles and specific mutations. Therefore, it would be important to know the metaplastic carcinoma subtypes of the PDX models used. In addition, for clinical application, it would be important to understand the effect of NOSi sensitization to chemotherapy according to the metaplastic carcinoma subtypes. For example, in ref 3, the squamous subtype was shown to have the highest expression of PI3K pathway proteins, which would suggest that NOSi may be more effective in this subtype.
2. Data in Figure 1 comes from an analysis of a previously published paper by the authors. Please

clarify what are the new data in this figure.

3. Figure 1E. The NOSi expression in the fibroblastic cells after treatment is high. Also, the images of the pAKT IHC staining appear non-specific. The quality of the IHC needs to be improved, to further inform the quantification (H-score) shown in Figure 1F.
4. Figure 2. The investigators evaluate the PDX models for expression of markers and mutation analyses. However, it would be important to also evaluate samples of human metaplastic carcinoma tissues (not PDX), as this is the primary tumor in the patient, and the PDX may be altered by passage in mice.
5. Figure 3. A rigorous synergistic assay with calculation of synergy score is needed.
6. The DNA damage response is not mechanistically investigated in detail.
7. Overall, quantification of WB is needed.
8. Figure 4. Was metastatic disease evaluated in the mice? There are no details on the histopathological changes induced by the various treatments in the PDX tumors.
9. The authors performed NOS KO in one cell line to investigate the mechanism. However, more than one KO or several shRNA KD need to be shown to ensure the specificity of the results
10. Figure 5. Panel B is not interpretable, which is quantified in C.
11. Figure 5E. 2D cell cultures show subtle morphological differences that were not quantified. Three dimensional organoid models or xenografts would be complement and enhance the 2D experiments.
12. The molecular connection with EMT could be studied in more detail (e.g., vimentin, N-cadherin, other EMT-TFs)
13. Figure 6H and Figure 7A are uninterpretable. The accompanying H&E needs to be provided for reference, as it is not possible to see which cells are expressing the markers, and in which cellular compartment (e.g., E-cadherin is expressed in the cell membranes, ALDH1 in the cytoplasm). Because Figure 7A is uninterpretable, the quantification of this stain is also unreliable. Further details are needed for the limiting dilution assays.
14. In the discussion the authors acknowledge that using PDX models preclude investigations on the tumor immune microenvironment. However, they state that there is a lack of available well-validated immunocompetent murine models Indeed, there have been several metaplastic breast carcinoma models reported in the literature (e.g., PMID 34508101, 27819674, and 26100884).

Response to Reviewers

Reviewer #1 - Breast cancer stemness, EMT, orthotopic models (Remarks to the Author):

Reddy et al. describe a possible “Achilles heel” of metaplastic breast cancer (MpBC) in the form of nitric oxide synthases (NOS). They could show that NOS inhibition increases the response to phosphoinositide 3-kinase (PI3K) inhibitor alpelisib as well as taxane in vitro and in vivo. The group goes on to show mechanistic evidence of the NOS blockade acting through repressing the JNK/c-Jun pathway. This likely increases chemosensitivity and is responsible for slower and less tumor growth in vivo. Reddy and colleagues investigate this effect in detail in cancer stem-like cells and show a reversal in epithelial-mesenchymal transition (EMT) as well as a decrease in cancer stem cell markers when investigating NOS-repressed MpBC cell lines. Overall this story makes for a very interesting read with a host of data supporting the authors model. The writing is concise and conclusive and does not over-interpret the data which is great to see these days. The manuscript would benefit greatly from a more thoroughly written methods section, there is some information missing (see below). In addition some experiments seem necessary to support the conclusions which I detailed below.

Major revision:

- Figure 1E, add PTEN and pAKT IHC for EOT

Response 1:

We thank reviewer 1 for this suggestion of adding PTEN and pAkt IHC to Figure 1E. We included those new images in a revised Figure 1D. To create more space in the main Figure 1, we also moved the original Figure 1B plot to supplemental Figure 1A and included a revision in the main manuscript in lines 184 and 187.

- Limiting dilution TP in 6F does not help to make the point. An ELDA analysis requires three not two points to generate a decent output. I suggest 100000, 10000 and 1000 cells for transplants

Response 2:

We thank Reviewer 1 for this suggestion regarding the limiting dilution assay result in Figure 6F. We have re-done this study but with a particular focus on the docetaxel treatment arm compared to triple combination (docetaxel+L-NMMA+alpelisib), as prior studies have shown that breast cancer stem cells are predominately chemoresistant and their population can be enriched post-chemotherapy (Phi, Sari et al. 2018), as indicated in reference #33 of the manuscript, and we wanted to evaluate whether combined PI3K and NOS inhibition could effectively target resistant breast cancer stem cells with tumor-initiating potential that may be enriched from chemotherapy. As suggested by the reviewer, we performed this analysis with 1000, 10000, and 100000 cells from each treatment arm that were implanted in NSG mice. We implanted these cells into the fat pad of 12-15 SCID/Beige mice, as indicated in the methods sections and evaluated for development of tumors in 12 weeks. Our new table is included in Figure 6F and the new results are discussed in lines 418-423, highlighted in yellow.

- 3E: Explain why the NOS Inhibitor L-NMMA has almost the exact same effect than the PI3K Inhibitor alpelisib on iNOS, it is difficult to then argue synergy. Maybe this should be phrased more careful

Response 3:

We thank Reviewer 1 for this comment and agree that in the previous immunoblots presented, it was difficult to fully assess whether L-NMMA and alpelisib were synergistic. We redid these

experiments by treating SUM159, Hs578T, and SUM159 cells for 24 hours with vehicle control, single-agent or combination therapy of L-NMMA and alpelisib for 24 hours. As you can see in the immunoblots after 24 hours of treatment in Figure 3E, NOS inhibition resulted in reduced phosphorylation of Akt and pS6 in MpBC cell lines SUM159 and Hs578T (line 290), but this effect was not seen in BT549 cell line. Furthermore, when we combined L-NMMA and alpelisib, we saw an even greater effect on reduced phosphorylation of Akt and pS6 in these two cell lines that actually resembled the synergistic effect of the two inhibitors. Furthermore, we performed this experiment three times and included the densitometry analysis in Supplemental Figure 3.

- 4F/G: it is hard to imagine, that the only difference between 4F and 4G is the PIK3CA mutation. The way I understand these models derived from patients with more than just one mutation. The conclusion that the response (which in itself is not that dramatic) is solely based on PIK3CA seems a little far-fetched.

Response 4:

We thank Reviewer 1 for this comment and after further examining our data, we also agree that it would be far-fetched to suggest that the difference in response is solely due to difference in *PIK3CA* mutation status. In fact, our data shows that while not significant, we do note a trend towards an augmented response of the PI3K inhibitor when combined with the NOS inhibitor, suggesting that this combination may be effective regardless of mutation status, which we indicated in lines 320-325 and in Figure 4E.

- 5F the E-cadherin change seems almost not present in the SUM159 KO cells, whereas HS578T changes in E-cadherin seem quite dramatic. So why not add more cell lines to the experiments in Figure 5, that would also add quality and reproducibility to Figure 5G now being derived from a single cell line.

Response 5:

We thank Reviewer 1 for this comment and added new data in Supplemental Figure 5M-O and lines 398-400 using Hs578T cells treated for 96 hours with siRNAs targeting *NOS2*, a non-targeting control, and parental Hs578T showing that genetic targeting of *NOS2* in this other cell line is associated with reduced phosphorylation of Akt, reversal of EMT, reduced activation of c-JUN, resulting in reduced protein expression of EMT effectors TGF β and LCN2 similar to the effect of pharmacological inhibition of *NOS2* on expression of E-cadherin (Fig. 5D). There are currently only four known breast cancer cell lines that have metaplastic pathology and based on our results from Figure 3, we only found two cell lines, SUM159 and Hs578T that are responsive to the combination therapy. As a result, we are limited in this study with adding more cell lines, but we still see a similar phenomenon of EMT reversal in the Hs578T cell line, as we did with SUM159.

- Figure 1F NR6 is unlikely to be non-significant, please correct like stated in the text

Response 6:

We thank Reviewer 1 for this comment and this figure has been corrected and it is now Figure 1E.

- Figure 1H and 1I are misrepresented in the text, please correct

Response 7:

We thank Reviewer 1 for this comment and have made the appropriate corrections in the text as indicated in yellow highlight in line 218 for Figure 1G and line 222 for Figure 1H.

- Line 223 seems to address Figure J on F

Response 8:

We thank Reviewer 1 for this comment and have made the appropriate corrections in the text as indicated in yellow highlight in line 224, which is now Figure 1I and Supplemental Figure 1C.

- 3B the figure is too small to identify the effect. Rather remove it, put a higher resolution in the supplementary files and leave the quantification in the main files.

Response 9:

We thank Reviewer 1 for this comment and have moved Figure 3B to Supplementary Figure 3C and have indicated this change in line 282.

- 6B is not a mammosphere assay when 60000 cells can cluster in a plate, good colony forming assays in Methylcellulose or Matrigel or Agar can identify the actual stem cell number in a given sample.

Response 10:

We thank Reviewer 1 for this comment regarding mammosphere assay, however the protocol that we used to evaluate mammosphere formation from patient-derived xenograft tissue was a well-established protocol that we use in the lab and was also published in Clinical Cancer Research (Schott, Landis et al. 2013), as referenced (#58) in the manuscript.

Why would the single docetaxel/L-NMMA and alpelisib treatment increase the number of CSCs in 6D?

Response 11:

We thank Reviewer 1 for this question, and we suspect that single-agent taxane therapy may be enriching a highly chemoresistant breast cancer stem population, as shown previously in other studies (Creighton, Li et al. 2009, Lu, Chen et al. 2017), indicated in reference #6 and #34 . Figure 6D show representative images of the flow cytometry analysis that was performed in triplicates for every treatment condition. Our data shown in Figure 6G suggests that single-agent L-NMMA and alpelisib may be effective at targeting a relatively smaller percentage of CSC, and their effect is even more prominent when we combine the PI3K and NOS inhibitor together.

- There is a discrepancy between representative images in 6D versus the evaluation in 6E.

Response 12:

We thank Reviewer 1 for this comment and the images in Figure 6D are representative images of our experiment in which we performed with 3-5 animals per treatment condition. The evaluation in 6E shows the mean of the % of CD44+/CD24- cells that were detected by flow cytometry.

- Reading total ALDH1 fluorescence is not an adequate test, please use the FACS analysis if you wish to show ALDH1 expressing cell changes

Response 13:

We thank Reviewer 1 for this comment and have added the new flow cytometry data for ALDH1+ cells in Figure 6G-H and moved the ALDH1 immunofluorescence data to supplemental Figure 6B-E.

- Method section does not adequately describe the xenografts in Figure 4 and Figure 6: cell number? Size of a tumor piece? More detail is required here.

Response 14:

We thank Reviewer 1 for this comment and have added more information regarding our xenograft studies in lines 644 and lines 661-681 regarding our breast cancer stem cell assays that were performed with results in Figure 6/Supplemental Figure 6.

Reviewer #2 - NOS signaling (Remarks to the Author):

Comments to the authors:

In the manuscript “NOS inhibition sensitizes metaplastic breast cancer to alpelisib and taxane chemotherapy by reversing c-JUN-mediated epithelial-to-mesenchymal transition,” the authors demonstrate that decreasing cellular levels of nitric oxide through inhibition of iNOS, which is overexpressed in many breast cancer subtypes with poor prognoses, may offer a method of treating disease. Specifically, they find that NOS inhibition reduces deleterious transcriptional changes promoted by JNK and c-Jun, leading to higher differentiation, decreased EMT, reduced cancer stem cell behavior, and enhanced sensitivity to chemotherapy modalities. In all, this work offers incremental improvement to existing therapy and proposes a molecular model through which this new approach may act.

Clarifications:

1. What exactly is the effect of iNOS on PI3K and how does this differ with PI3KCA mutation? Is the kinase S-nitrosylated?

Response 1:

We thank Reviewer 2 for this question. As we have discussed in our previous review article (Reddy, Glynn et al. 2023), via various S-nitrosation mechanisms, iNOS is capable of modulating PI3K/Akt signaling. In melanoma models, iNOS-derived NO can reversibly S-nitrosylate tuberous sclerosis 2 (TSC2) protein, impairing TSC2/TSC1 dimerization, resulting in mammalian target of rapamycin (mTOR) activation and enhanced proliferation of melanoma cells (Lopez-Rivera, Jayaraman et al. 2014). Furthermore, iNOS-derived NO can S-nitrosylate phosphatase and tensin homolog (PTEN) protein, thereby attenuating PTEN phosphatase activity and stimulating PI3K/Akt signaling (Ding, Ogata et al. 2021). Lastly, it has been shown in breast cancer cells that iNOS associated activation of Akt requires TIMP1 protein nitration and TIMP1 and CD63 protein-protein interactions. (Ridnour, Barasch et al. 2012). We discuss these examples in lines 523-530 and in references #17, #35, #45, #46 of the manuscript.

These listed mechanisms for how iNOS can activate PI3K signaling are all independent of *PIK3CA* mutation status and from our review of the literature, there is no current published evidence as to whether the kinase itself gets S-nitrosylated.

2. GSNOR expression (and SCOR/AKR1A1 if possible) should be assessed in cells and PDX. GSNOR is often downregulated in breast cancers and may help explain the variable response to NOS inhibition.

Response 2:

We thank Reviewer 2 for this comment and evaluated the expression of GSNOR in MpBC cell lines and found that in the cell lines that responded well to the combined inhibition of iNOS and PI3K, they tended to have relatively lower expression of GSNOR compared to cell lines that had did not respond to combined inhibition (BT549 and HCC1806). We added this result as Figure 3E and lines 292-297 in the manuscript.

Additional comments:

1. Attention to various details throughout the paper and consistency in representation should be made. A few examples:

- The p-value listed in the text for Figure 1I does not match that shown in the figure

Response 3:

We thank Reviewer 2 for this comment and the figure was mislabeled in the manuscript and it is now corrected to be Figure 1H and also in line 222.

- The text does not mention Figure 1J (it is listed as 1F instead)

Response 4:

We thank Reviewer 2 for this comment and checked that there is no Figure 1J in the main figure.

- There is not consistency on hyphenating L-NMMA and c-Jun, etc.

Response 5:

We thank Reviewer 2 for this comment and checked the manuscript to ensure that hyphenations for L-NMMA and c-Jun were correct. We made corrections for c-JUN as indicated in yellow highlight in lines 388, 391, and 393.

- Figure 6 text and legend does not seem to point to the correct data for labels of 6G-F (should be H)

Response 6:

We thank Reviewer 2 for this comment and made the appropriate corrections that are highlighted in yellow to indicate figure 6H in lines 427 in the manuscript text and line 985 in the figure legend.

- Figure legends are generally sparse and do not include details such as the colors of the bars and abbreviations used in Figure 7

Response 7:

We thank Reviewer 2 for this comment and revised all the figure legends, highlighted in yellow, to add more details regarding colors of bars in graphs, about abbreviations we used in figures, and general methodology relevant to the figure.

- Certain figures that include p-values above bars express the actual p-value, while others simply indicate “ $p < 0.05$.” An effort should be made to consistently show the true values atop all such bars.

Response 8:

We thank Reviewer 2 for this comment and have revised Figure 1E, Figure 6B-C, E, and Figure 7C-F to include all p-values for data that were previously indicated as non-significant (ns) and have checked the entire manuscript to ensure that all other data have actual p-values.

2. Figure 1F: it seems hard to believe that there is no significance between the points in sample NR6

Response 9:

We thank Reviewer 2 for this comment and this figure has been corrected and it is now Figure 1E and we also added p-values for the other columns that were previously indicated as non-significant (ns).

3. Figure 2B: please explain the relevance of correlated mutations in these pathways and how their enrichment in pan-cancer implicates an interplay between NOS and these respective pathways.

Response 10:

We thank Reviewer 2 for this comment and have included in lines 242-245 and lines 506-522 an explanation that co-occurrence of genomic alterations in NOS2 and PI3K signaling genes suggests that these two pathways may collaborate with each other to induce tumorigenesis in multiple cancer types by utilizing similar mechanisms.

4. Figure 2E: since iNOS is graded into buckets for relative expression, could the same be done for pAkt as well to make the correlation easier to visualize?

Response:

We thank Reviewer 2 for this suggestion and have revised figure 2E to include gradients of protein expression of pAkt for each PDX.

5. Figure 3C: there is significant leaky expression of iNOS in the iNOS knockout lines. Is the knockout only in a single allele?

Response:

As indicated in Figure 3C and 5F, in our SUM159 cells in which we targeted iNOS using the double Nickase iNOS CRISPR plasmids, we did not see substantial leaky expression of iNOS to justify single allele targeting. This is clearly manifested by increasing the exposure of our immunoblots that did not result in a corresponding increase in the band size in lanes with SUM159 KO protein lysate indicating an irrelevant background signal. Furthermore, in supplemental Figure 3D-E, we also compared the nitrite/nitrate concentrations in parental SUM159 and NOS2KO cells and saw a substantial reduction in production of nitrite/nitrate in NOS2KO cells.

6. Figure 3D: are there statistics that can be used to describe the difference between the curves? They are visually quite similar but may hold important differences.

Response:

We thank Reviewer 2 for this question and we performed statistical analysis (Kolmogorov-Smirnov test) on the NOS2KO curves compared to the parental SUM159 cells and our p-values were trending toward significance, but did not reach significance. We did add the calculated IC₅₀ concentration for each cell line and fold change differences in the IC₅₀ concentration of the NOS2KO cells compared to the parental cells.

7. Figure 3E: This figure was difficult to interpret and may benefit from some of the following modifications:

- Removal of the 4-hour data, since much of it does not show the concluded trends that are clear at 24 hours
- Quantification of Western blots with error bars for replicate experiments. This should be included in general for Western blots.
- Clarification on certain data: why L-NMMA treatment alone reduces iNOS expression significantly; why SUM cells do not have a reduction in pAkt for Thr (only Ser) but the other cells have reductions in both; why pAkt increases with L-NMMA alone at 24-hours; why SUM responds the worst with these readouts but seems to exhibit the best response for DSBs in the subsequent Figure 3F

Response:

We thank Reviewer 2 for this comment and redid the experiment, now shown in Figure 3E and quantification is in supplemental Figure 3. We show that in SUM159 and Hs578T cells, there is reduction in iNOS, pAkt, and pS6 expression that is prominent with combined PI3K and NOS inhibition. On the other hand, in the BT549 cell line that exhibit poor response to this combination therapy, we do not see that associated reduction in protein expression of iNOS, pAkt, and pS6. These immunoblots support our findings from the cell viability, comet assays, and crystal violet stain colony formation assays.

8. Figure 4A: the numbers under the day arrowheads are undefined. These may indicate weeks, but one of the clusters includes 8 arrowheads. Please clarify in figure legends.

Response:

We thank Reviewer 2 for this comment and have revised Figure 4A to indicate 1 and 2 are week 1 and week 2. We also added in the figure legend (line 911-912) the meaning of red and green triangles for each week and we hope this clarification reduces any further confusion in the figure.

9. Please mention the selection of particular cell lines or xenograft models used for certain experiments. For example, Figure 4B-E shows four models with varying degrees of response, while subsequent experiments use different groupings of two models.

Response:

We thank Reviewer 2 for this comment and included in lines 355-357 that we specifically wanted to investigate how EMT reversal occurred in our metaplastic cell line models by using cell lines that were responsive to combined therapy, which were SUM159 and Hs578T and this was based on the data we presented in Figure 3.

10. Figure 5B and other immunofluorescence: labels above the columns would make it easier to read the data, and indicating the merged column.

Response:

We thank Reviewer 2 for this comment and have revised the immunofluorescence images to label the columns so that it would be easier to read. These revisions are made in Figure 5B, and Supplemental Figure 5B and 6B, 6D.

11. Figure 5G: the significance threshold is listed as $p < 0.1$. Are these values corrected for multiple statistical tests? If not, perhaps a more stringent threshold could be assigned.

Response:

We thank Reviewer 2 for this comment and reviewed the data. The figure legend that had indicated $p < 0.1$ should be $p < 0.05$ and this has been corrected in lines 943 and 944. Figure 5G shows uses a threshold of p -value < 0.05 for upregulated and downregulated genes, that are indicated as red and green dots respectively.

12. Figure 5O: this experiment does not have any replicates with error bars and should be done at least three times independently. The SNO protein should be expressed as a densitometry ratio to the total JNK protein, which is first normalized to the HSP90 control, not as a ratio to the HSP90 control. The axis has a typo in the name of the HSP90 protein.

Response:

We thank Reviewer 2 for this comment, and we independently performed the S-nitrosylation three times and have revised the graph in Figure 5O to reflect a densitometry ratio of SNO-JNK/total JNK.

13. Figure 6B/C: the statistical comparisons in these figures are for each group compared to the untreated controls. If the case is to be made that the triple therapy is more beneficial than any of the double therapies, especially the double therapy that does not include L-NMMA, then a comparison between the triple therapy and the double therapies should be included as well. That the triple therapy is also different than the untreated control does not help to build a case for the inclusion of L-NMMA. At least the comparison between the triple therapy and the bar immediately to its left should be shown.

Response:

We thank Reviewer 2 for this comment and added p -values comparing triple therapy to both double therapies and we found that triple therapy significantly reduced secondary MSFE compared to both double therapies. We also added a description about the red and black p -values in lines 974-976 for figure legend of Figure 6.

14. Figure 7G: the figure legend and text of the paper describe roles for Akt that are not illustrated in the figure. Inclusion of something for Akt would help summarize the findings. We thank Reviewer 2 for this comment and have revised Figure 7G and lines 1003-1005 to include that EMT mediator TGF β can actually go back and activate PI3K/Akt signaling by stimulating Type I and II serine/threonine kinase receptor complex (T β RI/T β RII), causing T β RI to associate with p85, mediating Akt activation, inactivation of tuberous sclerosis complex, and eventually activate mTOR.(Zhang, Zhou et al. 2013), as referenced in the manuscript (#58).

15. Line 273 mentions a reverse trend (antagonistic) in two cell lines. An explanation of why this unexpected reversal in these cells would be helpful.

Response:

We thank Reviewer 2 for this comment and have added a few sentences in lines 462-470 in our discussion to further highlight this discussion point as to why we see antagonistic activity in BT549 and HCC1806 cell lines.

16. Line 303 uses the phrase NOSi and PI3Ki, which presumably mean “inhibition,” although this is not defined. NOSi may be confusing due to its similarity to iNOS.

Response:

We thank Reviewer 2 for this comment and made revisions in line 314, 315, 884, and 904 that are highlighted in yellow.

17. Lines 308-309, 341: the order mentioned in the text is different than the order mentioned in the figures

Response:

We thank Reviewer 2 for this comment have made the appropriate changes in the order of what is mentioned in text vs figures, and they are highlighted in yellow in lines 320-321 and 357.

18. Discussion: please provide a summary of the scope of this finding within the context of other cancers and other potential targets of iNOS inhibition. One such candidate is ovarian cancer, which may share some key similarities with estrogen and iNOS.

Response:

We thank Reviewer 2 for this comment and have added this summary in lines 506-522 in the discussion of the manuscript.

Reviewer #3 - Metaplastic breast cancer, sequencing (Remarks to the Author):

This is an interesting and exciting study demonstrating that NOS inhibition sensitizes metaplastic breast cancer to alpelisib and taxane chemotherapy and the underlying potential mechanisms, which involve reversal of EMT and stemness. The most important aspect of the study is the synergism of the drug combination that offers potential therapeutic strategy for the highly aggressive breast cancer subtype termed metaplastic carcinoma. Following are several comments that in my view would enhance the study.

1. Metaplastic breast carcinomas have several well-defined pathological subtypes, which are not really investigated in this study. A detailed recent proteogenomic study (ref 3) demonstrates that these subtypes have common as well as distinct proteomic profiles and specific mutations. Therefore, it would be important to know the metaplastic carcinoma subtypes of the PDX models used. In addition, for clinical application, it would be important to understand the effect of NOSi sensitization to chemotherapy according to the metaplastic carcinoma subtypes. For

example, in ref 3, the squamous subtype was shown to have the highest expression of PI3K pathway proteins, which would suggest that NOSi may be more effective in this subtype.

Response 1:

We thank Reviewer 3 for this comment and have added in lines 492-497 about the specific subtype metaplastic tumors in the PDXs that responded to combined NOS and PI3K inhibition.

2. Data in Figure 1 comes from an analysis of a previously published paper by the authors. Please clarify what are the new data in this figure.

Response 2:

We thank Reviewer 3 for this comment. The paper we are referencing was published in Science Translational Medicine in 2021 (Chung, Anand et al. 2021), referenced (#13) in the manuscript. All of the data presented in this submitted manuscript is novel and was previously not published/reviewed in any other journal.

3. Figure 1E. The NOSi expression in the fibroblastic cells after treatment is high. Also, the images of the pAKT IHC staining appear non-specific. The quality of the IHC needs to be improved, to further inform the quantification (H-score) shown in Figure 1F.

Response 3:

We thank Reviewer 3 for this comment and we re-did the staining for iNOS and pAkt to help improve the quality of the stain and these new photos are in Figure 1D.

4. Figure 2. The investigators evaluate the PDX models for expression of markers and mutation analyses. However, it would be important to also evaluate samples of human metaplastic carcinoma tissues (not PDX), as this is the primary tumor in the patient, and the PDX may be altered by passage in mice.

Response 4:

We thank Reviewer 3 for this comment. Figure 1A shows the data for iNOS staining at baseline for the metaplastic tumors at baseline and its relationship to tumor change from baseline post-treatment. Supplemental Figure 1B shows the available data for mutation analysis of 7/13 patients in our study. As indicated in line 204-205 of the manuscript, we did not have mutational analysis available for 6/13 patients in our study because either the analysis was not conducted or because the data was not available for us for further analysis. These patients were referred to our institute from an outside institution and unfortunately, we did not have all of the pertinent next generation sequencing of those 6 patients for further analysis.

5. Figure 3. A rigorous synergistic assay with calculation of synergy score is needed.

Response 5:

We thank Reviewer 3 for this comment and have included results of our cell viability assay and synergy/antagonism scores using the CalcuSyn software analysis in Figure 3A.

6. The DNA damage response is not mechanistically investigated in detail.

Response 6:

We thank Reviewer 3 for this comment and most of our DNA damage response studies are in Supplemental Figure 3 and Figure 3G (comet assay). The DNA damage response studies support the overarching hypothesis that NOS inhibition decreases cell stemness (5E and G) and induces EMT reversal, resulting in MpBC to become more sensitive to PI3K inhibition and taxane therapy. As previous studies have shown that PI3K inhibitors such as alpelisib have DNA damaging effects via nucleotide depletion (Juvekar, Hu et al. 2016), as referenced (#25) in the manuscript, we showed that NOS inhibition augments the DNA damaging effects of the

PI3K inhibitor and likely this may be attributed to the inhibitory effects of NOS blockade on cell stemness and EMT.

7. Overall, quantification of WB is needed.

Response 7:

We thank Reviewer 3 for this comment have added densitometry analysis of Figure 3E in Supplemental Figure 3 and revised image in Figure 5O.

8. Figure 4. Was metastatic disease evaluated in the mice? There are no details on the histopathological changes induced by the various treatments in the PDX tumors.

Response 8:

We thank Reviewer 3 for this question. Some of the PDXs we used in the study were from primary tumors from patients with metastatic disease, but when we evaluated the PDXs for metastatic disease in mice, we were unable to see any evidence of metastases (data not shown). Furthermore, we were unable to see any evidence of metastatic disease with MpBC cell lines that were injected in the mammary fat pad of Scid mice.

9. The authors performed NOS KO in one cell line to investigate the mechanism. However, more than one KO or several shRNA KD need to be shown to ensure the specificity of the results

Response 9:

We thank Reviewer 3 for this comment and added new data in Supplemental Figure 5M-O using Hs578T cells treated for 96 hours with siRNAs targeting NOS2, a non-targeting control, and parental Hs578T showing that genetic targeting of NOS2 in this other cell line is associated with reduced phosphorylation of Akt, reversal of EMT (E-cadherin, ZEB1), reduced activation of c-JUN, resulting in reduced protein expression of EMT effectors TGF β and LCN2 mimicking the effect of pharmacological inhibition of NOS on the EMT marker E-cadherin (5D). There are currently only four known breast cancer cell lines that have metaplastic pathology and based on our results from Figure 3, we only found two cell lines, SUM159 and Hs578T that are responsive to the combination therapy. As a result, we are limited in this study with adding more cell lines, but we still see a similar phenomenon of EMT reversal in the Hs578T cell line, as we did with SUM159.

10. Figure 5. Panel B is not interpretable, which is quantified in C.

Response 10:

We thank Reviewer 3 for this comment and the indicated images shown in Figure 5B are representative images of the immunofluorescence analysis we conducted, and the data is shown in Figure 5C.

11. Figure 5E. 2D cell cultures show subtle morphological differences that were not quantified. Three dimensional organoid models or xenografts would be complement and enhance the 2D experiments.

Response 11:

We thank Reviewer 3 for this comment. Figure 5E shows that the majority (about 80%) of parental cells grow as single spindle shaped cells (with length more than half of the white bar) compared with the NOS2KO cells that are more epithelial round shaped cells that their diameter does not exceed 1/5-1/7 of the white bar. Furthermore, we did not pursue organoid models in this study because we were concerned that organoids would be selective to highly chemoresistant CSC/de-differentiated cells and may not be reflective of the biology of MpBC. Utilizing organoid models would thus be out of scope for this study.

12. The molecular connection with EMT could be studied in more detail (e.g., vimentin, N-cadherin, other EMT-TFs)

Response 12:

We thank Reviewer 3 for this comment and in our study, we were able to show that in MpBC, nitric oxide via iNOS leads to S-nitrosylation of JNK, causing activation of c-JUN, and increased transcription of EMT effectors *LCN2* and *TGFB1*. In our supplemental studies, we showed that when we re-introduce TGF β back to NOS2KO cells, the mesenchymal features of these cells are restored and there is increased expression of Zeb1, Vimentin, Slug, and Snail TFs. These findings suggest that nitric oxide's relationship with EMT is via increased expression of EMT effectors *LCN2* and *TGFB1*.

13. Figure 6H and Figure 7A are uninterpretable. The accompanying H&E needs to be provided for reference, as it is not possible to see which cells are expressing the markers, and in which cellular compartment (e.g., E-cadherin is expressed in the cell membranes, ALDH1 in the cytoplasm). Because Figure 7A is uninterpretable, the quantification of this stain is also unreliable. Further details are needed for the limiting dilution assays.

Response 13:

We thank Reviewer 3 for this comment regarding the reliability of our immunofluorescence data. We added new flow cytometry data for ALDH1+ cells in Figure 6G-H and moved the ALDH1 immunofluorescence data to supplemental Figure 6B-E. As you can see with our new results, the flow cytometry data shows similar results to our immunofluorescence data for ALDH1. We also redid the LDA study but with a particular focus on the docetaxel treatment arm compared to triple combination (docetaxel+L-NMMA+apelisib), as prior studies have shown that breast cancer stem cells are predominately chemoresistant and their population can be enriched post-chemotherapy (Phi, Sari et al. 2018), as indicated in reference #33 in manuscript, and we wanted to evaluate whether combined PI3K and NOS inhibition could effectively target resistant breast cancer stem cells with tumor-initiating potential that may be enriched from chemotherapy. We performed this analysis with 1000, 10000, and 100000 cells from each treatment arm that were implanted in NSG mice. We implanted these cells into the fat pad of 12-15 SCID/Beige mice, as indicated in the methods sections and evaluated for development of tumors in 12 weeks. Our new table is included in Figure 6F and we placed the older LDA study results in Supplemental Figure 6A. We also provided accompanying H&E staining images in Supplementary Figure 7 and in lines 438-440 for responder 3 and nonresponders 1.

14. In the discussion the authors acknowledge that using PDX models preclude investigations on the tumor immune microenvironment. However, they state that there is a lack of available well-validated immunocompetent murine models Indeed, there have been several metaplastic breast carcinoma models reported in the literature (e.g., PMID 34508101, 27819674, and 26100884).

Response 14:

We thank Reviewer 3 for this comment. A few immunocompetent murine models have previously been reported but their appropriateness regarding their frequency of metaplastic tumor development, spontaneous tumor latency (5-14 months) and growth rate for our multiple combinatorial treatments is technically debatable. In the present study, we report and focus on strong tumor intrinsic effects of NOS and underlying mechanisms that correlate well with clinical responses. Studying the NOS inhibitor in tumor microenvironment will require a separate ambitious approach with development and analysis of functional syngeneic models out of these previously reported spontaneous tumors and this is the focus of our on-going and future studies. We included this description in lines 537-540 in the manuscript.

Reviewers' Comments:

Reviewer #1:

Remarks to the Author:

all comments were addresses more or less adequately. The manuscript is now suited for publication

Reviewer #2:

Remarks to the Author:

I think this is OK to be published pending minor revision. Specifically, the abstract (and text) would benefit from mechanistic detail. "NOS inhibition" is not sufficient.

Specifically, What does it mean to say "Mechanistically, NOS blockade directly acts on JNK/c-Jun complex to" How does 'blockade act on'?

NOS inhibition is not the same as kinase inhibition (the latter provides mechanistic insight into target, pathway and PTM). NOS blockade just means one inhibited NO production. Target/PTM is key! One can't develop drugs without knowing the target.

Presumably, the authors meanby inhibiting JNK S-nitrosylation. If so say so (and if not do the experiment to prove it).

Minor minor.

The authors refer to "S-nitrosation" --the chemical modification by NO+. But the chemical mechanism is not actually known in these instances. The neutral term S-nitrosylation, which refers to PTM rather than chemistry, is preferred.

Reviewer #4:

Remarks to the Author:

The authors present a substantially revised manuscript which fully addresses the majority of referee #3's points, apart from point 11. I am not sure I buy the argument why 3D cultures were not used. Surely 3D is more reflective of the in vivo situation? Chemoresistant/stem-like cells would likely only be seen in non-adherent spheres, not in organoids. This is still a weakness in the paper. However, as much of the work is done with PDXs, that mitigates the lack of in vitro 3D studies.

There are a few minor corrections which still need to be made:

Point 5: Details of the calcsyn software / method need to go into the methods section

Point 9: Use of only two cell lines is still not fantastic, however, the point that there are only two suitable ones available is very valid and needs to be mentioned in the text as mitigation

Point 10 and point 13: The authors may have missed the point of the referee's comment – the issue here for me is that while low magnification is necessary for representative quantification, high magnification is necessary so that individual cells can be seen and the staining judged to be actually in the cells, not non-specific stickiness. For each of the immunofluorescence figures, a high mag image of representative staining, showing 5 – 10 cells or so, should be provided as supplementary

Point 14: Although I think the authors are correct, the use of an immunocompetent GEMM model is

probably out of scope for the paper, considering the time and latency of the tumours in such models, a slightly fuller discussion of the literature around them would support their findings. In addition to the pubmeds already quoted, 24615332 shows squamous tumours (amongst others) develop in models in which Pten loss is targeted to the mammary epithelium, while 24615332 shows a direct correlation between pAKT levels and the squamous metaplastic phenotype, and proposes a model in which PI3K levels are a key determinant of tumour phenotype. A number of studies have also demonstrated the link between spindle cell metaplasia and resistance to olaparib in BRCA1/2 mouse models (eg 31080552, 26100884)

Response to Reviewers:

Reviewer #1 (Remarks to the Author):

all comments were addresses more or less adequately. The manuscript is now suited for publication

Reviewer #2 (Remarks to the Author):

I think this is OK to be published pending minor revision. Specifically, the abstract (and text) would benefit from mechanistic detail. "NOS inhibition" is not sufficient.

Comment 1:

Specificaly, What does it mean to say "Mechanistically, NOS blockade directly acts on JNK/c-Jun complex to" How does 'blockade act on'?

Response 1:

We thank Reviewer 2 for this comment. In lines 334-342, we indicated that S-nitrosylation of JNK results in activation of the protein (reference 31) and enhanced JNK-associated transcriptional output. In our study, we found that NOS inhibition results in lower levels of nitric oxide (NO) causing less S-nitrosylation of JNK, less expression of JNK-mediated phosphorylation of c-JUN at Serine 63 and 73, and reduced expression of EMT mediators, TGF β and lipocalin. In addition, we have updated the abstract to more accurately describe the mechanism of action of NOS inhibition.

Comment 2:

NOS inhibition is not the same as kinase inhibition (the latter provides mechanistic insight into target, pathway and PTM). NOS blockade just means one inhibited NO production. Target/PTM is key! One can't develop drugs without knowing the target.

Presumably, the authors meanby inhibiting JNK S-nitrosylation. If so say so (and if not do the experiment to prove it).

Response 2:

We thank Reviewer 2 for this comment. We have revised to include this point in lines 475-478 of the manuscript.

Comment 3:

The authors refer to "S-nitrosation" --the chemical modification by NO $^+$. But the chemical mechanism is not actually known in these instances. The neutral term S-nitrosylation, which refers to PTM rather than chemistry, is preferred.

Response 3:

We thank Reviewer 2 for this comment and made the recommended changes throughout the manuscript.

Reviewer #4 (Replacement for Reviewer #3) (Remarks to the Author):

The authors present a substantially revised manuscript which fully addresses the majority of referee #3's points, apart from point 11. I am not sure I buy the argument why 3D cultures were not used. Surely 3D is more reflective of the in vivo situation? Chemoresistant/stem-like cells would likely only be seen in non-adherent spheres, not in organoids. This is still a weakness in the paper. However, as much of the work is done with PDXs, that mitigates the lack of in vitro 3D studies.

There are a few minor corrections which still need to be made:

Comment 1:

Point 5: Details of the calcsyn software / method need to go into the methods section

Response 1:

We thank Reviewer 4 for this comment and have added these details in lines 544-552

Comment 2:

Point 9: Use of only two cell lines is still not fantastic, however, the point that there are only two suitable ones available is very valid and needs to be mentioned in the text as mitigation

Response 2:

We thank Reviewer 4 for this comment and as indicated in lines 299-301, we mentioned that we had to use Hs578T and SUM159 for the remainder of our studies because those were the two cell lines that showed synergistic response to PI3K and NOS inhibition. We also included in the discussion in lines 417-426 regarding our hypotheses as to why the other two cell lines (BT549 and HCC1806) were not responsive to this combination therapy.

Comment 3:

Point 10 and point 13: The authors may have missed the point of the referee's comment – the issue here for me is that while low magnification is necessary for representative quantification, high magnification is necessary so that individual cells can be seen and the staining judged to be actually in the cells, not non-specific stickiness. For each of the immunofluorescence figures, a high mag image of representative staining, showing 5 – 10 cells or so, should be provided as supplementary

Response 3:

We thank Reviewer 3 for this comment. However, unfortunately these immunofluorescence images were captured two years ago and we currently do not have high quality magnified images of the human and PDX tissues were had performed immunofluorescence analysis.

Comment 4:

Point 14: Although I think the authors are correct, the use of an immunocompetent GEMM model is probably out of scope for the paper, considering the time and latency of the tumours in such models, a slightly fuller discussion of the literature around them would support their findings. In addition to the pubmeds already quoted, 24615332 shows squamous tumours (amongst others) develop in models in which Pten loss is targeted to the mammary epithelium, while 24615332 shows a direct correlation between pAKT levels and the squamous metaplastic phenotype, and proposes a model in which PI3K levels are a key determinant of tumour

phenotype. A number of studies have also demonstrated the link between spindle cell metaplasia and resistance to olaparib in BRCA1/2 mouse models (eg 31080552, 26100884)

Response 4:

We thank Reviewer 4 for this comment and have added these references with a discussion in lines 496-505.